



# A three-dimensional variational data assimilation system for aerosol optical properties based on WRF-Chem: design, development, and application of assimilating Himawari-8 aerosol observations

5    Daichun Wang[1], Wei You[1], Zengliang Zang[1], Xiaobin Pan[1], Yiwen Hu[1,2], Yanfei Liang[1]

[1]Institute of Meteorology and Oceanography, National University of Defense Technology, Changsha, 410005, China

[2]School of Atmospheric Physics, Nanjing University of Information Science & Technology, Nanjing, 211101, China.

*Corresponding author: Wei You; Email: ywlx_1987@163.com





**Abstract.** This paper presents a three-dimensional variational (3DVAR) data assimilation (DA) system for aerosol optical properties, including aerosol optical depth (AOD) retrievals and lidar-based aerosol profiles, which was developed for the Model for Simulating Aerosol Interactions and Chemistry (MOSAIC) within the Weather Research and Forecasting model coupled to Chemistry (WRF-Chem) model. For computational efficiency, 32 model variables in the MOSAIC_4bin scheme are lumped into 20 aerosol state variables that are representative of mass concentrations in the DA system. To directly assimilate aerosol optical properties, an observation operator based on the Mie scattering theory was employed, which was obtained by simplifying the optical module in WRF-Chem. The tangent linear (TL) and adjoint (AD) operators were then established and passed the TL/AD sensitivity test. The Himawari-8 derived aerosol optical thickness (AOT) data were assimilated to validate the system and investigate the effects of assimilation on both AOT and $PM_{2.5}$ simulations. Two comparative experiments were performed with a cycle of 24 h from November 23 to 29, 2018, during which a heavy air pollution event occurred in North China. The DA performances of the model simulation were evaluated against independent aerosol observations, including the Aerosol Robotic Network (AERONET) AOT and surface $PM_{2.5}$ measurements. The results show that Himawari-8 AOT assimilation can significantly improve model AOT analyses and forecasts. Generally, the control experiments without assimilation seriously underestimated AOTs compared with observed values and were therefore unable to describe real aerosol pollution. The analysis fields closer to observations improved AOT simulations, indicating that the system successfully assimilated AOT observations into the model. In terms of statistical metrics, assimilating Himawari-8 AOTs only limitedly improved $PM_{2.5}$ analyses in the inner simulation domain (D02); however, the positive



effect can last for over 24 h. Assimilation effectively enlarged the underestimated $PM_{2.5}$
concentrations to be closer to the real distribution in North China, which is of great value
for studying heavy air pollution events.

**Key words:** Aerosol optical properties; Data assimilation; WRF-Chem; 3DVAR; $PM_{2.5}$
forecasts

## 1. Introduction

Atmospheric aerosols have considerable impacts on weather, climate, and human health.
They are involved in many physical and chemical processes in the atmosphere, such as
directly scattering and absorbing solar radiation, sources of cloud condensation nuclei,
and air pollution (Pöschl, 2005; Gao et al., 2015; Chen et al., 2019). Conventional
observations such as surface mass concentration measurements play an important role in
aerosol analysis and monitoring, for instance, China has established a nationwide
monitoring network consisting of more than 1500 stations since 2013 to provide near
real-time data of pollutants, including $PM_{2.5}$, $PM_{10}$, $SO_2$, $NO_2$, CO, and $O_3$. However,
conventional observations alone are insufficient to describe three-dimensional aerosol
distribution in detail, because monitoring stations are mostly located in urban areas and
aerosol profiles, which are important for studying aerosol transport, are scarce. Light
extinction is an inherent property of aerosols, and different aerosol particles have
different extinctions, so optical observations are utilized to study aerosols. Compared
with conventional observations, optical properties can cover a much larger domain and
provide detailed aerosol profiles (Kaufman et al., 2002), which are bound to extend the
aerosol study. Furthermore, with the development of remote sensing technology, more





aerosol optical properties have become available. For example, Moderate Resolution Imaging Spectroradiometer (MODIS) aerosol optical depth or thickness (AOD or AOT) data have been widely used (Liu et al., 2011; Schwartz et al., 2012; Saide et al., 2013), Aerosol Robotic Network (AERONET; http://aeronet.gsfc.nasa.gov/) (Holben et al.,

1998) AOD observations have been used for aerosol analyses (Rubin et al., 2017; Dai et al., 2019), and AOT retrievals from the Japanese Himawari-8, a next-generation geostationary meteorological satellite, have been operationally used since 2015 (Sekiyama et al., 2016). Additionally, aerosol extinction or backscattering coefficient detected by ground-based lidar (Wang et al., 2014; Cheng et al., 2019) or space-borne

lidar such as Cloud-Aerosol Lidar and Infrared Pathfinder Satellite Observation (CALIPSO) has also been employed to analyze aerosol profiles (Sekiyama et al., 2010). Besides, numerical simulations conducted by atmospheric chemistry models or air quality models have increasingly played an essential role in aerosol analysis and prediction. Significant progress has been achieved in recent years; however, accurate

aerosol modeling remains challenging given the large uncertainties associated with aerosol emissions, initial conditions, and complex interactions with meteorological processes. Solving these uncertainties is of great significance for improving aerosol modeling. Data assimilation (DA), a statistically optimal approach combining observations with numerical model outputs, can reduce uncertainties in the initial aerosol

fields. Chemical DA, especially aerosol DA, has gradually developed to improve the prediction of air quality in recent years. In early studies of aerosol DA, the optimal interpolation (OI) technique was employed to assimilate the total mass concentrations of $PM_{2.5}$ or $PM_{10}$ using a control variable scheme (Tombette et al., 2009). The variational algorithm was also employed in some studies. Niu et al. (2008) used the three-



dimensional variational (3DVAR) technique to assimilate dust aerosol observations based on one control variable and obtained a positive assimilation result. With an understanding of the aerosol chemical mechanism, as well as the improvement of computing performance, multi-variable aerosol DA studies were conducted, which mainly focused on the development of the 3DVAR technique and Coupled Chemistry Meteorology Model (CCMM). For example, the open-source Grid-point Statistical Interpolation (GSI) tool presented by National Centers for Environmental Prediction (NCEP) (Wu et al., 2002; Kleist et al., 2009) has been widely applied to aerosol DA (Pagowski et al., 2010; Liu et al., 2011; Jiang et al., 2013; Feng et al., 2018;Pang et al., 2018). The GSI tool was preliminary developed for the Goddard Chemistry Aerosol Radiation and Transport (GOCART) aerosol scheme (Chin et al., 2000) using the 3DVAR algorithm. To overcome the systematic underestimation of the GOCART scheme in the assimilation context, researchers developed an aerosol DA system based on the Model for Simulating Aerosol Interactions and Chemistry (MOSAIC) aerosol scheme (Li et al., 2013; Zang et al., 2016; Wang et al., 2020; Liang et al., 2020). For instance, Li et al. (2013) lumped eight aerosol species within MOSAIC into five control variables and then constructed a 3DVAR DA system to assimilate $PM_{2.5}$ mass concentrations, and the results showed that DA has a beneficial effect on both the initial field and $PM_{2.5}$ forecasts within a 24 h period. Although the four-dimensional variational (4DVAR) technique has been extensively used in meteorological operations (Gauthier et al., 2007), it has only been employed to assimilate atmospheric chemical compositions such as $O_3$, $SO_2$, and CO based on a simple offline chemical transport model (CTM) because of the high computational cost and complex adjoint model (Eibern and Schmidt,





1999; Elbern and Schmidt, 2001). Consequently, the 3DVAR algorithm is still commonly used for aerosol DA.

As mentioned above, optical properties have great potential for studying aerosols, so it is natural to incorporate them into models via assimilation. The key issue of directly assimilating aerosol optical properties is the establishment of an observation operator and its adjoint for variational methods. Liu et al. (2011) added the forward AOD operator and its adjoint module within the Community Radiative Transfer Model (CRTM) (Han et al.,

2006) to the GSI for the first time and successfully assimilated MODIS AODs. This extended assimilation tool was then employed to assimilate various AOD retrievals from different platforms (Schwartz et al., 2012; Saide et al., 2014; Tang et al., 2017; Pang et al., 2018; Ha et al., 2020), and achieved encouraging results. Similar to AOD, assimilating lidar aerosol profiles also involves the complex forward operator and its

adjoint (Cheng et al., 2019; Wang et al., 2014). In order to simplify the observation operator, an approximate approach was utilized to directly assimilate aerosol profiles. For example, Liang et al. (2020) employed the Interagency Monitoring of Protected Visual Environments (IMPROVE) equation which is the linear link between the extinction coefficient and aerosol chemical species mass as the forward operator to

construct a 3DAVR DA system and then assimilated ground-based lidar aerosol profiles and $PM_{2.5}$ mass concentrations simultaneously. Also, some researchers have used sequential approaches, such as the ensemble Kalman filter, to advance aerosol DA (Schutgens et al., 2010; Yumimoto et al., 2016; Sekiyama et al., 2016; Dai et al., 2019). Nevertheless, the ensemble based aerosol forecasts are very expensive duo to the heavy

computational load, especially online meteorology-chemistry modelling, it is difficult to widely implement them in the operational air quality DA systems (Pang et al., 2021).





Following Li et al. (2013) and You (2017), this study further extends the assimilation of aerosol optical properties. Using an observation operator based on the Mie scattering theory, a comprehensive 3DVAR DA system aiming for aerosol optical properties,
including AOD retrievals and aerosol profiles, is developed for the MOSAIC aerosol scheme within the Weather Research and Forecasting model coupled to Chemistry (WRF-Chem) model for the first time. The remainder of this paper is organized as follows. Sect. 2 presents the aerosol DA system in detail. The data and experimental methods used in this study are described in Sect. 3. The background error statistics
necessary for the assimilation experiment are analysed in Sect. 4. The results are summarized in Sect. 5, discussing the assimilation effects. Finally, a summary is presented in Sect. 6, along with discussions on the limitations of this study and suggestions for future research.

## 2 Aerosol data assimilation design

### 2.1 Model description

WRF-Chem is an advanced online coupled meteorology-aerosol model (Grell et al., 2005) that can simultaneously simulate meteorological fields and atmospheric chemical compositions including aerosols. It has been widely used in air quality forecasting and aerosol-related studies (Chen et al., 2016). Aerosol processes are treated by modules or
schemes in WRF-Chem, such as GOCART (Chin et al., 2000), MOSAIC (Zaveri et al., 2008), and Modal Aerosol Dynamics Model for Europe (MADE) (Ackermann et al., 1998). There is no size information but total mass for sulfate, black carbon (BC), and organic carbon (OC), while there is size information only for dust and sea salt in





GOCART, in addition to no description of second organic aerosol (SOA), resulting in its numerical efficiency. Due to more detailed descriptions of dust, GOCART has been applied more extensively in dust aerosol research. MADE is a modal aerosol scheme that describes more aerosol species than GOCART, including sulfate, ammonium salt, back carbon, organic carbon, sea salt, nitrate, dust, and SOA. Moreover, it employs three log-normal modes, that is, Aitken, accumulation, and coarse, to describe aerosol size distributions in detail. Although such a scheme is ideal for aerosols, it consumes more computational resources; therefore, its applications are limited. As a newly developed scheme, MOSAIC is a sectional aerosol scheme that incorporates tradeoffs between detailed descriptions of aerosol chemical species, size distributions, and computational cost. Previous studies have shown that this scheme has a good ability to simulate the compound aerosol pollution process in China (Gao et al., 2015; Chen et al., 2016; Chen et al., 2019). The MOSAIC scheme divides atmospheric aerosols into eight species, including BC, OC, nitrate ($NO_3^-$), sulfate ($SO_4^{2-}$), chloride ($Cl^-$), sodium ($NA^+$), ammonium salt ($NH4^+$), and other unclassified inorganic mass (OIN). At the same time, 4 or 8 discrete size sections or bins are employed to represent the size distribution of each species. In this study, we selected four bins for computational efficiency. The first, second, third, and fourth size sections are set to be 0.039–0.1 μm, 0.1–1.0 μm, 1.0–2.5 μm, and 2.5–10.0 μm, respectively. The sum of the eight species in the first three sections corresponds to $PM_{2.5}$, whereas the sum of all the sections corresponds to $PM_{10}$. This approach ensures that aerosols are represented efficiently and accurately. Thus, it can be concluded that the MOSAIC aerosol mechanism of multiple species in multi-particle size sections has an advantage in anthropogenic aerosol studies over other





schemes. Therefore, we conducted aerosol analyses and forecasts using MOSAIC, and a DA system was developed for the MOSAIC scheme.

The WRF-Chem version 4.0 was used to perform assimilation simulation experiments. Both physical and chemical parameterization schemes are indispensable for numerical simulations. The main parameterization schemes used in this study include the WRF single-moment 6-class microphysics scheme (WSM6, Hong and Lim, 2006), the Rapid Radiative Transfer Model for General Circulation Model (RRTMG) longwave and shortwave radiation scheme (Iacono et al., 2008), the Noah land-surface scheme (Chen and Dudhia, 2001), the Yonsei University (YSU) atmospheric boundary layer scheme ( Hong and Lim, 2006), the Grell-Freitas convective parameterization scheme, the Fast-J photolysis scheme (Ruggaber et al., 1994), the Regional Acid Deposition Model, version 2 (RADM2, Stockwell et al., 1990)/MADE/Second Organic Aerosol Model (SORGAM, Schell et al., 2001) anthropogenic emissions, and the MOSAIC_4bin scheme described above.

The configuration of the two-level nested simulation domain is shown in Fig. 1a, including most of East Asia in Domain 1 (denoted by D01, hereafter) with a horizontal grid spacing of 27 km, and the entirety of North China as well as parts of East and Central China in Domain 2 (denoted by D02, hereafter) with a horizontal resolution of 9 km, 1/3 that of D01. To ensure a detailed simulation of aerosol vertical distributions, 40 vertical layers were modelled in the simulation, and it is worth mentioning that the vertical resolution was not set to be uniform but decreased with height. The lowest layer is at the surface, whereas the top reaches 50 hPa.





## 2.2 Basic formulation

The 3DVAR algorithm has been extensively used for aerosol analysis and forecasts, such as the GSI tool, because of its high computational efficiency and the advantages of handling unconventional observations. Thus, it was employed to construct a DA system aiming for aerosol optical properties in this study. The 3DVAR algorithm can produce an aerosol analysis field with minimum analysis error covariance after a correction to the

background field through the introduction of various observation information. For this purpose, an incremental approach was adopted, similar to the operational use in meteorology (Courtier et al., 1998). In its incremental formulation, 3DVAR attempts to minimize the objective function $J$.

$$J(\delta\mathbf{x}) = \frac{1}{2}\delta\mathbf{x}^{\mathrm{T}}\mathbf{B}^{-1}\delta\mathbf{x} + \frac{1}{2}(\mathbf{H}\delta\mathbf{x} - \mathbf{d})^{\mathrm{T}}\mathbf{R}^{-1}(\mathbf{H}\delta\mathbf{x} - \mathbf{d}) \tag{1}$$

where $\delta\mathbf{x}$ is the increment, corresponding to an aerosol state vector that defines the state variables of three-dimensional grid, also known as control variables in the DA process. At the minimum, the resulting analysis increment $\delta\mathbf{x}^{\mathrm{a}}$ is added to the background $\mathbf{x}^{\mathrm{b}}$ to provide the analysis $\mathbf{x}^{\mathrm{a}}$. $\mathbf{B}$ is the background error covariance matrix, and $\mathbf{d}$ is the innovation vector, which is expressed as:

$$\mathbf{d} = \mathbf{y} - H[\mathbf{x}^{\mathrm{b}}] \tag{2}$$

where $\mathbf{y}$ is the observation vector. $\mathbf{H}$ is a suitable linear approximation of the observation operator $H$ in the vicinity of $\mathbf{x}^{\mathrm{b}}$, known as the tangent linear (TL) operator, and its transpose is the adjoint (AD) operator (see below). $\mathbf{R}$ is the observation error covariance matrix. Sect. 5 describes the calculation of $\mathbf{B}$. In most cases, observations are

independently conducted, so $\mathbf{R}$ is assumed to be a diagonal matrix without correlations among different observation errors considered. In general, observation errors associated



with AOD retrievals are determined by measuring instruments. According to Yumimoto et al. (2016), the observation error of Himawari-8 AOT retrievals is set to 0.06 in this study. However, further studies on observation errors of aerosol optical properties are necessary.


The search for a minimum solution to the objective function usually involves a numerical iterative process using a descent algorithm. However, it is difficult to solve using Eq. (1) because it includes the inverse of **B**. We used the methods of Li et al. (2013) to deal with the inversion of **B**. First, **B** can be represented as the product of submatrices (Bannister, 2008), $\mathbf{B}=\mathbf{DCD}^{\mathrm{T}}$, where **D** is the background error standard deviation (STD) matrix, and **C** is the background error correlation matrix. Second, a Cholesky factorization is applied to **C** because it is a symmetric and positive definite matrix. The Cholesky factorization is:


$$\mathbf{C} = \mathbf{C}^{1/2}\left(\mathbf{C}^{1/2}\right)^{\mathrm{T}} \tag{3}$$

where the matrix $\mathbf{C}^{1/2}$ is a lower triangular matrix. Using this Cholesky factorization, we can transform the control variables $\delta\mathbf{x}$ to $\delta\mathbf{z}$ through:


$$\delta\mathbf{x} = \mathbf{DC}^{1/2}\delta\mathbf{z} \tag{4}$$

Finally, substituting Eq. (4) into Eq. (1), we obtain the desired form of Eq. (1) as

$$J(\delta\mathbf{z}) = \frac{1}{2}\delta\mathbf{z}^{\mathrm{T}}\delta\mathbf{z} + \frac{1}{2}\left(\mathbf{HDC}^{1/2}\delta\mathbf{z} - \mathbf{d}\right)^{\mathrm{T}}\mathbf{R}^{-1}\left(\mathbf{HDC}^{1/2}\delta\mathbf{z} - \mathbf{d}\right). \tag{5}$$

The transformed objective function is generally better conditioned, and thus, this transformation expedites convergence when it is iteratively minimized. Along with the objective function $J(\delta\mathbf{z})$ computed at each iterative step, the derivative of $J(\delta\mathbf{z})$ with respect to $\delta\mathbf{z}$ is computed as:


$$\nabla J(\delta\mathbf{z}) = \delta\mathbf{z} + \left(\mathbf{DC}^{1/2}\right)^{\mathrm{T}}\mathbf{H}^{\mathrm{T}}\mathbf{R}^{-1}(\mathbf{HDC}^{1/2}\delta\mathbf{z} - \mathbf{d}) \tag{6}$$





The iteration starts with δ**z** = 0 and does not finish until the convergence condition is met
or it reaches the maximum number of iterations. The descent algorithm used is the
limited memory BEGS method (LBFGS) (Liu and Nocedal, 1989). Finally, a return from
the resulting δ**z** at the minimum to δ**x**$^a$ is obtained through Eq. (4).

## 2.3 Control variables

As discussed above, the basic framework of Li et al. (2013) was employed to develop a
DA system. To assimilate aerosol optical properties, a set of control variables different
from those of Li et al. (2013) was designed, which are key elements in the DA system.
The control variables vary with the aerosol scheme. Because the background field **x**$^b$ was
simulated with the MOSAIC_4bin scheme within WRF-Chem described in Sect. 2.1, the
control variables should be designed according to the MOSAIC aerosol scheme. A total
of 32 model variables represent the mass concentrations of eight species in the four bins
within MOSAIC. If these model variables are directly taken as control variables, the
resulting increments directly correspond to the model variables and can be added to the
background to produce an analysis without intermediate conversions. However, such a
number of control variables, much more than those in meteorological DA, will cause a
heavy burden on computational and memory resources and even lead to computational
non-convergence when the cost function is iteratively minimized. Therefore, a reduction
in the model variables is essential for a stable and efficient assimilation system, meaning
that the model variables should be lumped into fewer control variables in the DA process.
The control variables generally depend on the available observations to be assimilated.
For example, when assimilating routine aerosol measurements, such as the total mass
concentrations of PM$_{2.5}$, and PM$_{10}$, the lumped control variables represent the total mass





concentrations of different aerosol species without the size information included. For instance, the four model variables for sulfate so4_a01, so4_a02, so4_a03, and so4_a04 were reduced to two control variables: one was the sum of so4_a01, so4_a02, and so4_a03 and the other was so4_a04 itself (Wang et al., 2020). However, for aerosol optical properties, the size information must be reserved within the control variables because aerosol particles with different size distributions have significantly different light extinctions. Nevertheless, some species have similar optical characteristics, including density and complex refractive indices, such as sulfate ($SO_4^{2-}$), nitrate ($NO_3^-$), and ammonium salt ($NH4^+$). Thus, we lumped these species so that the species treated by the assimilation system were reduced to black carton, organic carton, the summation of sulfate, nitrate, and ammonium salt, the summation of chlorides and sodium salts, which are quite rare inland, and other unclassified inorganics, and these five species were denoted by EC, OC, SSN, CN, and OIN, respectively (You, 2017), the size information of which was retained using the same four bins as in Sect. 2.1. Consequently, there are a total of 20 control variables, named after EC1, EC2, EC3, EC4, OC1, OC2, OC3, OC4, SSN1, SSN2, SSN3, SSN4, CN1, CN2, CN3, CN4, OIN1, OIN2, OIN3, and OIN4, where the numbers 1, 2, 3, and 4 represent the four size sections, respectively. These control variables can easily be obtained from the model variables and represent the mass concentrations of the five aerosol species within the four bins. It should be noted that the direct result of assimilating is to generate the increments of 20 control variables above here, and the increments of lumped variables should be distributed into individual model variable within MOSAIC. For instance, the increment of SSN1 is equal to the summation of that of the model variables so4_a01, no3_a01, and nh4_a01. For simplicity, the distribution ratio was determined using the mass concentration background error STD for





each model variable. When the increment of each model variable is obtained, directly adding that to its background value will produce an aerosol analysis.

## 2.4 Observation operator and its adjoint

The observation operator transforms the control variables into an equivalent of each observed quantity at the observation locations. Thus, a comparison between the simulations and observations can be performed, on which the resulting increments depend. A nonlinear operator based on the Mie scattering theory was employed to directly assimilate aerosol optical properties. Specifically, the 20 control variables described in Sect. 2.3 are used to compute optical parameters, such as aerosol extinction coefficient at every model grid point, and then both horizontal and vertical interpolations of the simulations from the model grid to observation locations are performed. This approach ensures that the simulated and observed quantities are comparable to each other. The process of the forward observation operator is to compute aerosol optical properties through the control variables, as shown in Fig. 2, based on the work of Wang et al. (2014) and Barnard et al. (2010).

Although computing aerosol optical properties with WRF-Chem outputs involves many aerosol variables, as shown in Fig. 2, for simplicity, only mass concentrations measured routinely were set as the control variables. The forward operator is composed of several steps as follows. First, it is assumed that aerosol chemical species are internally mixed along with water in each bin. Given the densities of five assimilated species as well as water, individual volume is easily obtained so that the mean wet radius $r_i$ assigned to each bin can be computed by dividing the total volume by the number concentration $N_i$, assuming that the particles are spherical, where the subscript $i$ denotes the size bin. The





particle size parameter is of significant importance to optical properties and is
determined by $x=2\pi r_i/\lambda$, where $\lambda$ is the incident wavelength. Each species is associated
with a complex index of refraction, and while these indices depend on $\lambda$, they vary little
in the short-wave range where aerosols are remotely measured. The indices at a
wavelength of 550 nm were therefore used to compute the averaged refractive index $m_i$
for each size bin by means of volume averaging for simplicity, which was reduced from
the optical properties (OP) module in WRF-Chem that employs the RRTMG scheme to
compute a set of refractive indices in the range of both long and short waves. It is worth
noting that the incident wavelength $\lambda$ is set as an input parameter according to aerosol
retrievals so that the size parameter can be accurately computed. Second, when the mean
wet radius $r_i$ and complex refractive index $m_i$ are given, optical efficiencies such as
extinction efficiency $Q_{ext}$ and scattering efficiency $Q_{sca}$ can be obtained through Mie
calculations, and this step is greatly crucial to the whole forward operator. Because Mie
calculation involves the operations of a complex variable, it is very difficult to establish
the computing codes and their adjoint codes. Fortunately, some Mie calculation modules
have been successfully developed by previous researchers and have been widely applied
in related studies. Among these modules, the routines provided by Wiscombe (1979)
behave perfectly in terms of computational stability and efficiency, so we can obtain
optical efficiencies at each grid point of three dimension by repeatedly calling it within a
loop. This approach ensures that optical efficiencies are accurately calculated; however,
this requires more computation time owing to complex nonlinear operations of Mie
scattering, and developing its adjoint is faced with great challenges and difficulties,
which will have an adverse impact on operational use. The methodology described by
Ghan et al. (2001) was used to efficiently calculate optical efficiencies by the OP module





in WRF-Chem (Fast et al., 2006). It employs a Chebychev polynomial expansion to fit extinction efficiency, absorption efficiency, scattering efficiency, asymmetry factor, and

backscattering efficiency, based on a sample generated from Mie calculations; for example, $Q_{ext}$ can be given by

$$Q_{ext} = \exp\left[\sum_{i=1}^{M} A_i T_i(s)\right] \tag{7}$$

where $s = (2\log r_i - \log r_{max} - \log r_{min})/(\log r_{max} - \log r_{min})$, a logarithm of the wet radius $r_i$, and both $r_{max}$ and $r_{min}$ are known parameters. $T_i(s)$ is the Chebychev polynomial

of order $i$, and $M$ is the number of terms in the expansion. The coefficients $A_i$ depend on the averaged refractive index for the size bin in question, and they are found using bilinear interpolation over a set of stored coefficients. Once $A_i$ is obtained, $Q_{ext}$ is easily computed using Eq. (7). This method is fast and results in maximum errors of just a few percent. More details regarding this methodology can be found in Fast et al. (2006).

Similarly, optical efficiencies are determined by the wet radius $r_i$ and refractive index $m_i$ as well as the wavelength $\lambda$ more efficiently than the Mie calculation, which also reduces the difficulties of developing the DA system. Hence, we directly transported the OP module to construct the forward observation operator. To perform efficiently, some routine codes unnecessary for the assimilation system were removed so that the forward

codes were dramatically reduced compared to the OP module, which is convenient for establishing the TL and adjoint codes. Finally, optical properties are determined by summation across all four size bins. For example, the extinction coefficient is given by:

$$b_{ext} = \sum_{i=1}^{4bins} N_i \pi r_i^2 Q_{ext}(r_i, m_i, \lambda) \tag{8}$$

and AOD is the column integration of $b_{ext}$ over the vertical layers. Obviously, the optical

properties simulated by the forward operator are distributed over the three-dimensional grid points. To directly compare the observations, spatial interpolation is needed.





As mentioned in Sect. 2.2, the TL and AD operators are used to compute the cost function and its derivative with respect to the control variables, respectively. Source-code transformation based on the chain rule is usually used to construct TL and adjoint codes, which is an augmentation of forward operator codes that have been already established and tested. Adjoint coding involves strict rules (Zou et al., 1997; Giering and Kaminski, 1998) and is also a heavy task if completed manually. The TL and adjoint codes were generated using the automatic differentiation tool TAPENADE V.3.15 (Hascoët and Pascual, 2013), which is available at http://www-tapenade.inria.fr:8080/tapenade/index.jsp. If a source program and its independent input variables and dependent output variables are given, the tool can generate the TL and adjoint programs, easing the burden of hand coding. The generated TL and adjoint codes were examined to ensure that they were correct prior to real application, and they passed TL/AD sensitivity test; for more details on how to check the TL and adjoint codes, please refer to Zou et al. (1997). Manual interventions are required when these generated codes are incorporated into the DA system, especially in the case of variable calculations on three-dimensional grid points. In addition, because the optical parameters are computed independently at each point, the forward, TL, and adjoint codes are properly organized in a parallel mode to further reduce the computation time.

With the increase in aerosol observations, the simultaneous assimilation of aerosol observations from various platforms has become a trend. This can be achieved by adding the summation associated with the corresponding observation items to the second term in the cost function described by Eq. (1). For this purpose, the system was developed to assimilate as many aerosol measurements as possible so that it has more potential for aerosol analysis and forecasting. In contrast to aerosol optical properties, assimilating



mass concentrations is elementary and easily performed using only a simple linear operator. The system developed here can assimilate optical properties, including extinction coefficient, backscattering coefficient, AOD, and even total attenuated backscattering coefficient (Sekiyama et al., 2010), and mass concentrations, including

total $PM_{2.5}$ or $PM_{10}$ mass and individual chemical species mass, simultaneously or separately, with a rational introduction of desired observational data, making it possible for further study.

## 3 Data and methods

Two comparative experiments were performed to assess the performance of assimilating

aerosol optical properties, which have the same model configurations described in Sect. 2.1, and the spin-up time was 24 h. The only difference between them is in the initial aerosol field. One is the reference experiment without any observations assimilated, simply taking the previous 24-h aerosol forecasts as an initialization, referred to as Control, while the other takes the aerosol analysis after assimilating satellite-derived

AOD as an initialization to simulate their subsequent variations, referred to as Assimilation. Both experiments employed the final (FNL) Operational Global Analysis data at a resolution of 1°×1° and 6 h interval presented by the National Centers for Environmental Prediction (NCEP) to generate the initial and lateral boundary conditions of the meteorological fields. The 2017 Multi-resolution Emission Inventory for China

(MEIC) collated by Tsinghua University (Zheng et al., 2018) was used for simulation. The experiment period started on November 23 and ended November 29, 2018, lasting one week, with a cycle of 24 h, during which an aerosol episode occurred in North China and considerable observational data were available.




The Himawari-8 AOT product was selected for assimilation by this system because it has

a much higher temporal coverage than that of polar-orbiting satellites, which is promising for aerosol DA, and has also been successfully assimilated using other methods (Sekiyama et al., 2016; Yumimoto et al., 2016; Dai et al., 2019). The Himawari-8 level 2 AOT is retrieved at 500 nm with a 10-minute observation interval as well as a 0.05° spatial resolution, however the data is noticeably noisy. The level 3 AOT,

including AOT_Pure and AOT_Merged, an improved hourly product, is an optimal estimation of AOT at a certain time rather than an estimate of the average state over an hour. AOT_Pure is a subset of level 2 AOT with strict quality control of cloud contamination, and AOT_Merged is the spatial and temporal optimum interpolation of AOT_Pure within an hour (Kikuchi et al., 2018). In this study, we focused on

assimilating the latest version of the Himawari-8 level 3 AOT_Merged at 500 nm, which contains as many AOT retrievals as possible with a horizontal resolution of 0.05°×0.05°. The original AOT data is commonly thinned before directly assimilating to avoid seriously overestimated increments caused by the much higher spatial resolution of AOT data than that of the model (Yumimoto et al., 2016; Dai et al., 2019; Ha et al., 2020).

Similar to Ha et al. (2020), we thinned the original AOT data over the D01 mesh (27 km) and D02 mesh (9 km), respectively, using the mean value of all the data points in one grid cell. A case of thinned AOTs retrieved at 0300 UTC on November 25, 2018, in D02 is shown in Fig. 3b, the number of the data is 13100 with a maximum value of 1.801. The AOT observations represent a heavy aerosol pollution episode that occurred in North

China, yet there is a lack of aerosol information in some heavily polluted regions due to cloud contamination, meaning that optical retrievals alone are not sufficient to thoroughly study aerosols. The Himawari-8 AOT is retrieved in the visible and near-





infrared bands, so the observation coverage differs with time of day. Nevertheless, the observations at 0300 UTC can nearly cover the whole of China, except some western areas. Hence, we chose the 0300 UTC rather than 0000 UTC, as used in usual experiments, as the initial time to perform a 24-h prediction of aerosols for the purpose of research.

To evaluate the performance of Himawari-8 AOT assimilation, three common statistical metrics, including the correlation coefficient (CORR), root mean squared error (RMSE), and mean bias (BIAS), were utilized (Boylan and Russell, 2006). It should be noted that compared with observations is the WRF-Chem D02 simulation, the results given below were computed using D02 outputs. First, we investigated the effects of assimilation on AOT simulations using assimilated Himawari-8 AOTs and independent observations, including MODIS AOD and AERONET AOT observations. Second, we investigated the effects of assimilating AOTs on $PM_{2.5}$ analysis and forecasting using hourly surface mass concentration observations (Fig. 1b) released by the China National Environmental Monitoring Centre (CNEMC). For instance, the $PM_{2.5}$ mass concentration observed at 0300 UTC on November 25, 2018, in D02 is shown in Fig. 3a, indicating a severe pollution zone in North China, which is largely consistent with the spatial representation of Himawari-8 AOTs.

## 4 Statistics of background error covariance

Background error covariance is an important issue in data assimilation, which not only specifies the spread of observation information in the background field, namely the way in which the observations affect the background values, but also determines the relative weight of observational and background information across the analysis field. In practice,





however, the error covariance **B**-matrix is too large for a multi-variable aerosol DA to be calculated numerically. For instance, the number of D02 grid points used here is in the order of $10^6$, in addition to 20 state variables, the number of elements in **B** is, therefore, $10^7 \times 10^7$. This size will result in difficulty for computing and storing **B**, therefore a simplification of **B** is required. Following the studies of Bannister (2008) and Li et al. (2013), the **B**-matrix was reduced to background error STD **D**, horizontal correlation matrix, and vertical correlation matrix, which can be computed separately. These three submatrices have dramatically fewer dimensions than **B**, so they become computationally treatable. Because the forecast error is unknown, most studies use model outputs to statistically estimate error covariance via modeling or parameterization, such as the NMC method (Parrish and Derber, 1992), which has been regularly used to calculate background error covariance for traditional meteorological fields such as temperature and wind and is also appropriate for aerosol mass concentrations (Benedetti and Fisher, 2007; Liu et al., 2011; Li et al., 2013). This study also utilizes the NMC method to calculate background error STDs, horizontal correlation, and vertical correlation based on differences between 48 h and 24 h forecasts valid at the same time (i.e., 0000 UTC) within a period of one month (November 2018). Because each aerosol state variable has a different background error covariance from others, which has been demonstrated by error statistics (see below), there is a need to estimate the error covariance for each variable to achieve better assimilation performance.

The error STD **D**-matrix of each variable is diagonal and was directly estimated as a domain average at every model level using WRF-Chem D01 and D02 outputs, respectively, and its vertical distribution (only for D02) is shown in Fig. 4. These STDs differ among aerosol variables. In terms of values, SSN2, SSN3, OIN2, and OIN4 have



larger error STDs than the others, with SSN2 having the largest value. The background error STDs are related to the aerosol species mass concentration. In general, variables with higher mass concentrations tend to have larger error STDs. For example, the simulation domain is far from the sea, and sea salt aerosols are very rare. As a result, no matter which size bin, the species CN has a significantly low error STDs below 0.05 μg m$^{-3}$, which is much lower than the other variables. These error STDs display a relatively rapid decrease with height apart from SNN2 and SNN3, but diminishing rates vary among aerosol variables. The fine structures of the error STD vertical distribution are related to the boundary layer heights. There is a noticeable increase in the SNN2 and SNN3 error STDs at the boundary layer height (approximately 1000 m).

The horizontal correlation matrix determines the propagation of observation information from the observation site to the surrounding area in the horizontal direction. Similar to Li et al. (2013), we assumed that different aerosol variables are not correlated; therefore, only auto-correlations of one variable at different distances were taken into consideration. For further simplification, we assumed that horizontal correlations are isotropic (Kahnert et al., 2008), which means that horizontal correlations are just a function of distance and have nothing to do with direction. Consequently, the horizontal correlation can be fitted using a one-dimensional Gaussian function. The correlation between two arbitrary points $x_1$ and $x_2$ can be expressed as $c(x_1, x_2)=\exp[-(x_1-x_2)^2/2L_v^2]$, where $L_v$ is the only unknown parameter, and is the horizontal correlation length scale of each state variable. The correlation increases as the distance decreases, especially when the distance decreases to zero, it obtains a maximum of 1. Thus, $L_v$ is defined as the distance at which the correlation decreases to $e^{-1/2}$ and can be calculated via model outputs. This distance averaged over the model domain was used as an estimate of $L_v$. The introduction of $L_v$



reduces the relatively complex two-dimensional correlation matrix to a parameter that is
able to completely describe the structure of horizontal correlation, undoubtedly
simplifying the computing and storage of the horizontal correlation matrix. The
estimated $L_v$ in D02 for individual aerosol state variable is given in Table 1. The
estimated correlation length scales are significantly different among the distinct species.
Thus, out of all aerosol variables, SSN3 has the largest scale at 47 km, indicating that the
influence of SSN3 observations could spread farther than other variables and has a
larger-domain improvement across the background field. In contrast, CN3 has the
smallest scale (12.8 km) and spreads the least based on observational information.
Overall, species SSN have relatively larger correlation length scales among species of
the same size section, except for in the fourth bin. Additionally, the same aerosol species
in different size sections have distinctly different error correlation length scales; for
example, OIN3 has a larger scale than OIN2. Such differences among the correlation
length scales indicate the need to use multi-species concentrations within the four size
bins as control variables.

Background error vertical correlation plays an important role in the vertical spread of
aerosol observation information. On the one hand, it has more complicated structures
instead of isotropy compared to the horizontal correlation because of the discontinuity-
like transition of the vertical distributions between the boundary layer and the free
atmosphere above, and such structures are difficult to represent using an analytic
function. On the other hand, the vertical correlation, which is the $nz \times nz$ (here, $nz$ is equal
to 40) matrix, is much smaller than the horizontal correlation matrix. As a result, the
vertical correlation was directly estimated using model outputs. Because the vertical
correlation of every variable is similar, the computed vertical correlations only for





control variables in the third size bin are shown in Fig. 5. A salient and common feature of these vertical correlations is that they decrease with height and have strong relation to the boundary layer heights, which means that aerosols are mainly stacked in the boundary layer and tend to accumulate closer to the ground. At the same time, consistent with the horizontal correlations, vertical correlations differ among aerosol variables. SSN3 has a relatively large vertical scale, whereas CN3 has a relatively small vertical scale, which is consistent with the horizontal features.

## 5 Results

### 5.1 Effects on AOT simulations

AOT is of great value for studying aerosol activities, which can be simulated by the forward operator within the DA system. In general, assimilating AOT certainly improves its analysis according to the basic principle of the 3DVAR algorithm, unless it is not successfully assimilated. It is noted that the wavelength variable necessary for computing AOTs described in Sect. 2.4 was set to be 500 nm, the wavelength at which the Himawari-8 AOTs are retrieved. A comparison between the simulated AOTs in the background field and analysis is usually employed to demonstrate the positive effects of assimilation. For illustration, the simulated AOTs as well as the so-called AOT increments at an initialization of 0300 UTC on November 25, 2018, are shown in Fig. 6. The increments, which are differences between the analysis and the background field, can be considered as the improvements generated by assimilation, including magnitude and range, and these increments are spatially consistent with the observations, which means that the observations have an important effect on the assimilation results.





Obviously, the simulated AOTs in the background field are dramatically underestimated
(Fig. 6a) compared with the observed Himawari-8 AOTs (Fig. 3b), while the analysis
brings the AOTs closer to the observations, which is indicated by the prominently
positive increments (Fig. 6c). At the same time, assimilation also decreases the AOTs
over other regions, with negative increments marked in blue. The background field is
570 generally unable to describe the real pollution, especially in the case of heavy pollution;
however, the analysis after assimilation can provide a relatively accurate pollution
situation (Fig. 6b).

The distributions shown in Fig. 6 express the effects of assimilating AOT on its analysis.
To quantitatively evaluate the effects, the three metrics described above, CORR, RMSE,
and BIAS, were computed through all the data pairs between the simulated and observed
AOTs after spatial interpolation from regular grid points to the corresponding
observational locations. The larger and closer to 1 CORR, the better the assimilation
performance, while the smaller RMSE and the closer to 0 BIAS, the better the
assimilation performance. Besides, the assimilated Himawari-8 AOTs were used to
580 compute the metrics, and another independent observation MODIS AOD was employed
to fully evaluate the effects of assimilation on the analysis. The Terra MODIS level 2
AOD data (MOD04_L2) were used for validation in this study. As this polar-orbiting
satellite passes over the equator at 10:30 local time, we collected all the data between
0000 UTC and 0600 UTC, rather than at a given time, to obtain more observations,
matching the simulated values at the initial time (i.e., 0300 UTC). It is worth mentioning
that MODIS AOD is retrieved at 550 nm and the simulated AOT is at 500 nm, which
will pose some but not largely significant effects on the evaluation. The experiment
lasted consecutively for a week in a cycle of 24 h, which contained seven initializations,



so we gathered the simulated AOTs at all the initializations to achieve a general evaluation result. The comparisons between the observed and simulated AOTs are presented using scatter plots, as shown in Fig. 7, where Fig. 7a represents the comparison with Himawari-8, and Fig. 7b shows a comparison with MODIS. The comparison with Himawari-8 AOTs reveals that the analyses have a better performance as CORR increases from 0.524 to 0.868, RMSE decreases from 0.280 to 0.147, and BIAS increases from $-0.133$ to $-0.031$ after assimilation. Similar results are found in the comparison with the MODIS AOTs. Red points are distributed denser and more parallel to the 1:1 line than the blue points, indicating that the analyses are closer to the observations. All three metrics demonstrated positive effects from assimilation on the analysis. In summary, the assimilation system can successfully introduce AOT observations into the model to generate a more accurate initial field.

Similar to other studies (Dai et al., 2019; Ha et al., 2020), an independent validation of the simulated hourly AOTs from both the control and assimilation experiments was conducted through a comparison with AERONET observations to further investigate the effects of assimilation on forecasting. There are a total of six AERONET sites in D02: Beijing, Beijing-CAMS, Beijing_PKU, Beijing_RADI, XiangHe, and XuZhou-CUMT, which are marked with red triangles in Fig. 1b. The sites can provide various AOT retrievals at different wavelengths, and those at 500 nm were selected for validation. In this study, we used level 2.0 and 1.5 (if level 2.0 data are not available) AERONET AOTs, which are cloud screened (Smirnov et al., 2000) and used to evaluate satellite observations. Fig. 8 depicts the time series of the simulated AOTs and observations at six AERONET sites from November 23 to 30, 2018. Compared with observations, the control experiment dramatically underestimated AOTs at all sites, while the assimilation





experiment significantly enlarged AOTs so that they became closer to the observations. This indicates that assimilation significantly improves AOT simulation. As can be seen, the assimilation benefits vary with sites; for instance, assimilation improves the AOT simulation at XuZhou-CUMT less than that at other sites (Fig. 8f), as well as the forecasting time; for example, the assimilation benefits for analyses can reach 24 h in the case of November 25, while they last less than 24 h in the case of November 24. The available observations largely account for this variation. A high pollution event took place on 26 November in North China so that AOTs over 1.6 were measured in Beijing (Fig. 8b), which can also be demonstrated by ground-level $PM_{2.5}$ observations (not shown here), but there is few Himawari-8 observations for the event to be assimilated due to cloud contamination. As a result, the assimilation experiment had the same performance as the control experiment, which is unable to describe the high pollution event. It has been concluded that the introduction of AOT observations by assimilation is beneficial to capture heavy pollution levels (Rubin et al., 2017).

## 5.2 Effects on $PM_{2.5}$ simulations

$PM_{2.5}$ mass concentrations draw large attention from both the public and researchers. They can be directly modeled using WRF-Chem and are conventionally measured at ambient air quality monitoring stations. As the DA system was developed based on the MOSAIC scheme, it should hopefully improve aerosol analyses and subsequent forecasts, especially for $PM_{2.5}$. North China is located in D02 and is known for its high levels of air pollution; therefore, WRF-Chem D02 outputs were directly employed to investigate the effects of assimilating Himawari-8 AOTs on regional $PM_{2.5}$ forecasts.





As described in Sect. 2.3, the assimilation process will produce the increments of 20 control variables. Of course, we can analyse every increment to assess the effects of AOT assimilation on the corresponding aerosol species simulations. Because there is a lack of observations for aerosol species at each size section, the total increment of $PM_{2.5}$ is analysed instead, which is simply a summation of increments over five assimilated

species in the first, second, and third size bins. For illustration, Fig. 9 only shows the simulated surface $PM_{2.5}$ concentrations in the background field and corresponding analyses at an initialization of 0300 UTC on November 25, 2018, as well as the increments between analyses and the background field. The control experiment underestimated $PM_{2.5}$ concentrations in North China compared with the observed values

(Fig. 3a). For example, the $PM_{2.5}$ concentrations in Tianjin reached more than 200 μg m$^{-3}$ while the simulated values in the background field were less than 150 μg m$^{-3}$ (Fig. 9a). The evidently positive increments generated by assimilation enlarge $PM_{2.5}$ analyses (Fig. 9c), making them closer to the observations, and the analyses are therefore able to describe heavy pollution. At the same time, negative increments decrease overestimation

in some places. The $PM_{2.5}$ increments are spatially consistent with AOT observations (Fig. 3b), which means that aerosol optical properties have been transformed into mass concentrations using the observation operator and then incorporated into the model. The analyses are superior to the background field in terms of pollution magnitude; however, the heavy pollution band in North China was simulated further to the east compared with

the observations. This might be ascribed to model deficiency in the representation of three-dimensional aerosol species.

Assimilation directly aims to improve aerosol analyses. As shown in Fig. 10, the data dots between simulated and observed $PM_{2.5}$ concentrations were also analysed according





to the three metrics. From Fig. 10, red points, standing for analyses, do not have a

significantly better performance than their blue counterparts for the control experiment, yet the metrics demonstrate the slight positive effects of AOT assimilation on aerosol analyses, increasing CORR from 0.485 to 0.530, decreasing RMSE from 60.66 µg m$^{-3}$ to 56.40 µg m$^{-3}$, and increasing BIAS from $-21.10$ µg m$^{-3}$ to $-16.13$ µg m$^{-3}$. This improvement is less significant than that of directly assimilating PM$_{2.5}$ concentrations

(Wang et al., 2020), however, the use of PM$_{2.5}$ concentrations to evaluate the effects of AOT assimilation is not objective and comprehensive because there is a discrepancy between PM$_{2.5}$ and AOT observations. For example, no assimilation benefits in some highly polluted areas are generated because of the lack of AOT retrievals, so the PM$_{2.5}$ observations cannot reflect the benefits from AOT assimilation.

To investigate the effects of AOT assimilation on PM$_{2.5}$ forecasts, time series of three metrics regarding the forecast range (i.e., 24 h) were computed using hourly WRF-Chem D02 outputs and observations. As shown in Fig. 11, in terms of both CORR and RMSE, the assimilation experiment performed better than the control experiment, indicating that the benefits for analyses from AOT assimilation can last up to 24 h. It is noted that the

assimilation benefits vary with integration time, decreasing in a fluctuating manner. The computed BIAS indicates that AOT assimilation improves PM$_{2.5}$ forecasts within 24 h, but can vary for certain times. As discussed above, assimilation significantly enlarges the simulated PM$_{2.5}$ concentrations, but an overcorrection, namely, the simulated values surpass observations, occurs approximately 7-8 h from the initial time (Fig. 11c), which

may be ascribed to the dramatically noisy AOT retrievals, as well as an imperfection of the observation operator for aerosol optical properties.





The time series of the simulated PM$_{2.5}$ concentrations and observations during the entire experimental period are shown in Fig. 12, which are hourly averaged over 683 stations in D02. The blue line denotes the control experiment, while the red line denotes the assimilation experiment, and the observations are represented using the black line. Overall, the mean PM$_{2.5}$ concentrations simulated by the assimilation experiment were closer to the observations than the control experiment, which is beneficial for describing the real heavy pollution in North China. Statistically, the CORR, RMSE, and BIAS between the black curve and blue curve were 0.645, 20.74 μg m$^{-3}$, and $-16.25$ μg m$^{-3}$ while CORR, RMSE, and BIAS between the black curve and the red curve were 0.732, 15.12 μg m$^{-3}$, and $-9.81$ μg m$^{-3}$, respectively, which means that the assimilation experiment had a better performance in PM$_{2.5}$ forecasts than the control experiment. These metrics indicate that AOT assimilation improves regional PM$_{2.5}$ forecasts, especially in the case of heavy pollution.

## 6 Summary and discussions

A 3DVAR DA system was independently developed to directly assimilate aerosol optical properties. This system was built based on the framework of Li et al. (2013) and developed for the MOSAIC scheme within WRF-Chem, a sophisticated aerosol model, rather than the GOCART scheme employed by CRTM. MOSAIC divides aerosol particles into eight species that are described in four size bins so that there are 32 mass concentration model variables. For computational efficiency, the 32 model variables were lumped into 20 aerosol state variables, which are representative of the mass concentrations of five assimilated species within the four size bins. An optical module



was added to assimilate aerosol optical properties, which consisted of the forward observation operator and its TL and AD codes. We properly reduced the OP module (Fast et al., 2006) in WRF-Chem to establish the forward operator, then the TL and AD codes were generated using an automatic differentiation tool and tested to ensure that they were correct. The system can assimilate aerosol optical properties such as extinction coefficient profile, AOD, and mass concentrations, simultaneously or separately, and these should be applied for further studies in the future.

Himawari-8 AOTs were assimilated to validate the system and investigate the effects of assimilation on both AOT and $PM_{2.5}$ simulations. A heavy air pollution event occurred in North China from November 23 to 29, 2018; therefore, this period was chosen for the simulation experiment. Two comparative experiments with a spin-up time of 24 h were performed, continuously lasting for a week with a cycle of 24 h. The control experiment took the previous 24 h aerosol forecasts as an initialization, while the assimilation experiment employed analyses after assimilating Himawari-8 AOTs to initialize the simulations. WRF-Chem D02 outputs were compared with the assimilated AOTs, independent MODIS AODs, AERONET AOT observations, and surface $PM_{2.5}$ mass concentration observations, respectively.

Background error statistics, including SDs, horizontal correlation length scales, and vertical correlations of 20 control variables, were estimated using monthly WRF-Chem outputs based on the NMC method, which are also necessary for the assimilation process. Our results showed that background error statistics distinctly vary among these control variables, which also illustrates the necessity of building a multi-variable aerosol DA system.

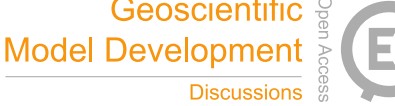

Assimilation significantly improves AOT analyses and forecasts. In general, the control experiment without assimilation seriously underestimated AOTs compared with the observed values. The analyses perform better in terms of the statistical metrics CORR, RMSE, and BIAS in comparison with both assimilated and independent AOTs than the background field. The analyses closer to observations improve AOT simulations, which is of great value in the study of AOT distribution during high pollution events. The improvement in AOT simulations indicates that the system successfully assimilated AOT observations into the model to form an accurate initial field.

Subject to the basic formulation, the DA process directly aims to improve aerosol analyses. In terms of statistical metrics, assimilating Himawari-8 AOTs improves $PM_{2.5}$ analyses, but not significantly, in D02 and the assimilation benefits can last more than 24 h. Assimilation significantly enlarges the underestimated $PM_{2.5}$ concentrations to be closer to the real distribution in North China during heavy pollution. The averaged surface $PM_{2.5}$ concentrations over D02 were better simulated during the whole pollution period after assimilation compared with corresponding observations, which means that AOT assimilation improves regional $PM_{2.5}$ simulations.

In this study, the observation errors of AOT retrievals were simply set as a constant. However, they should be determined by the retrieval uncertainty, or should be variable at least. Additionally, different thinning schemes for AOT retrievals may have different results. Consequently, these questions should be studied further. As more aerosol optical property observations become available, combined assimilation of optical properties and routine observations, such as aerosol extinction profiles and mass concentrations, has become popular. As described above, the system developed in this study has great



potential for assimilating various observations. Assimilating AOTs here is a preliminary study, and combined assimilation studies should be performed in the future.

*Code and data availability.* The WRF-Chem model source code can be downloaded at the WRF model download page (https://www2.mmm.ucar.edu/wrf/users/download/get_source.html, last access: 1 may 2021), The 3DVAR system was developed by the
authors. The exact version of the aerosol DA code and input data for supporting this paper are available at: https://doi.org/10.5281/zenodo.5528505.

*Author contributions.* DW and WY performed the assimilation experiments, prepared the dataset, upgraded data assimilation codes, performed the simulations, and drafted the manuscript. WY developed the 3DVAR data assimilation system, designed this study,
supervised the project of development, and revised the manuscript. All the authors continuously discussed the 3DVAR system development and the results of the manuscript.

*Competing interests.* The authors declare that they have no conflict of interest.

*Acknowledgments.* We are grateful to the National Centers for Environmental Prediction
for providing the FNL Operational Global Analysis data, China National Environmental Monitoring Centre for releasing pollutant observational data, NASA for providing MODIS AOD data, and the Japan Meteorological Agency for releasing Himawari-8 AOT observations. The AERONET Principal Investigators are also thanked for making their data available. This work was supported by the National Natural Science



Foundation of China (Grant Nos. 41805092 and 41775123) and the National Key R&D
Program of China (Grant No. 2017YFC0209803).

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



**Table 1. Horizontal correlation length scales for individual aerosol state variable**

| variable | EC1 | OC1 | SSN1 | CN1 | OIN1 |
|---|---|---|---|---|---|
| $L_V$ (km) | 29.2 | 30.2 | 31.6 | 20.9 | 26.5 |
| variable | EC2 | OC2 | SSN2 | CN2 | OIN2 |
| $L_V$ (km) | 36.4 | 38.4 | 43.3 | 20.3 | 32.7 |
| variable | EC3 | OC3 | SSN3 | CN3 | OIN3 |
| $L_V$ (km) | 41.0 | 42.5 | 47.0 | 12.8 | 37.4 |
| variable | EC4 | OC4 | SSN4 | CN4 | OIN4 |
| $L_V$ (km) | 37.3 | 38.0 | 35.0 | 14.6 | 25.5 |










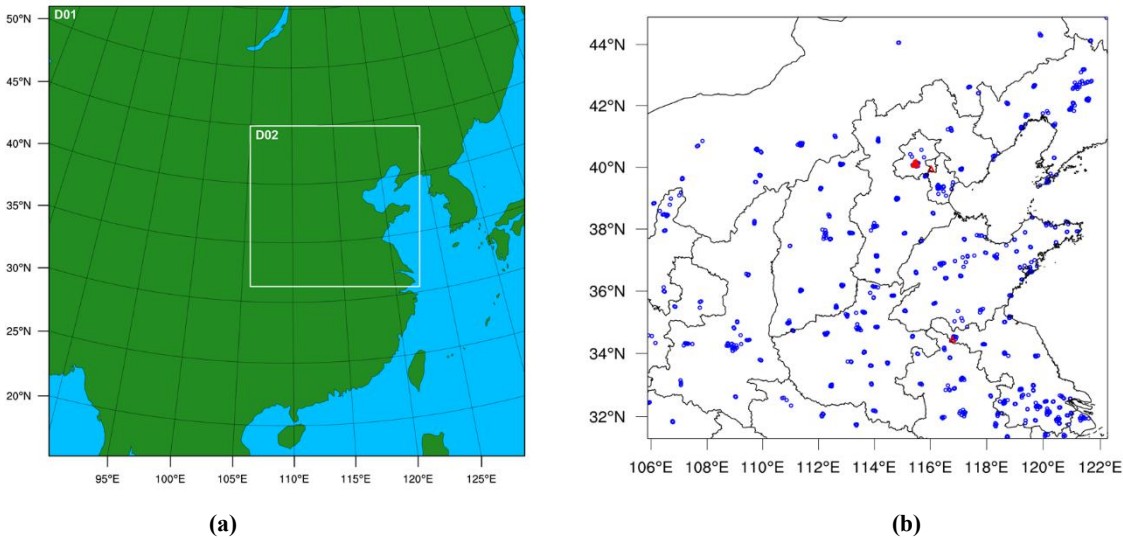

(a)                                                                (b)

**Figure 1. Configuration of the two-level nested domain used in this study (a) and the monitoring stations in Domain 2 (D02) (b). There are a total of 683 surface ambient air quality monitoring stations represented by little blue circles, which are mainly located in urban areas, as well as 6 AERONET sites represented by red triangles in D02. Both maps are plotted with NCAR Command Language Version 6.6.2.**









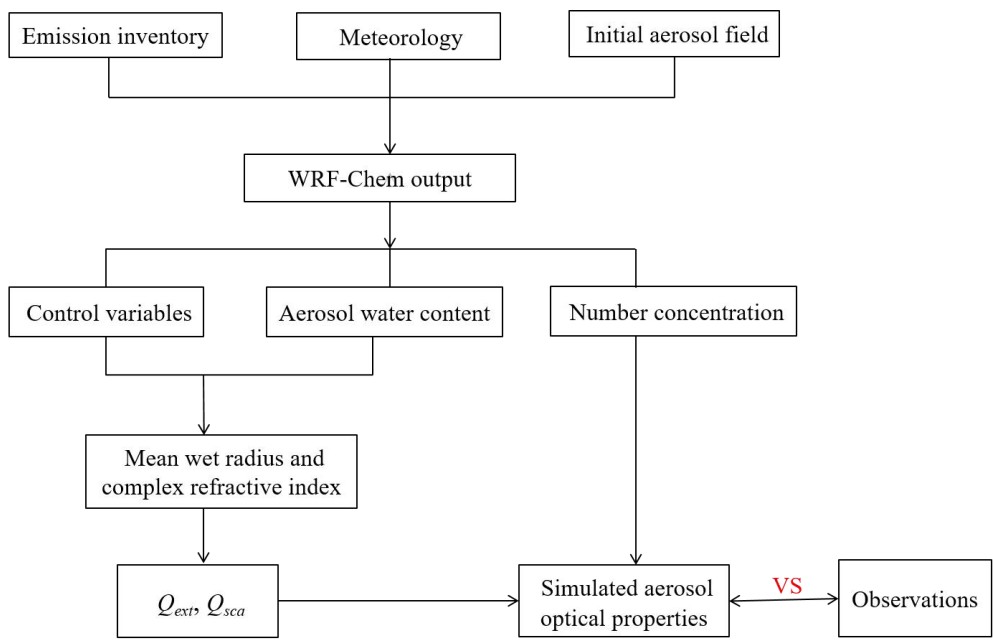

**Figure 2. Diagram describing the forward observation operator used to transform aerosol mass concentrations to optical parameters** *Qext* **and** *Qsca* **are extinction and scattering efficiencies respectively, which are functions of the size parameter and complex refractive index.**






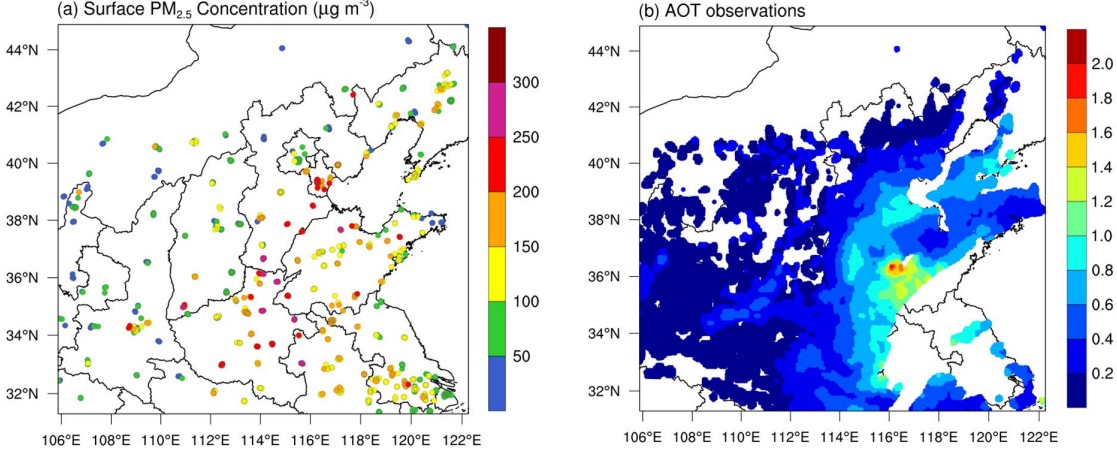

**Figure 3. Observations of surface PM$_{2.5}$ mass concentration (a) and the thinned Himawari-8 AOTs (b) in D02 at 0300 UTC on 25 November 2018.**









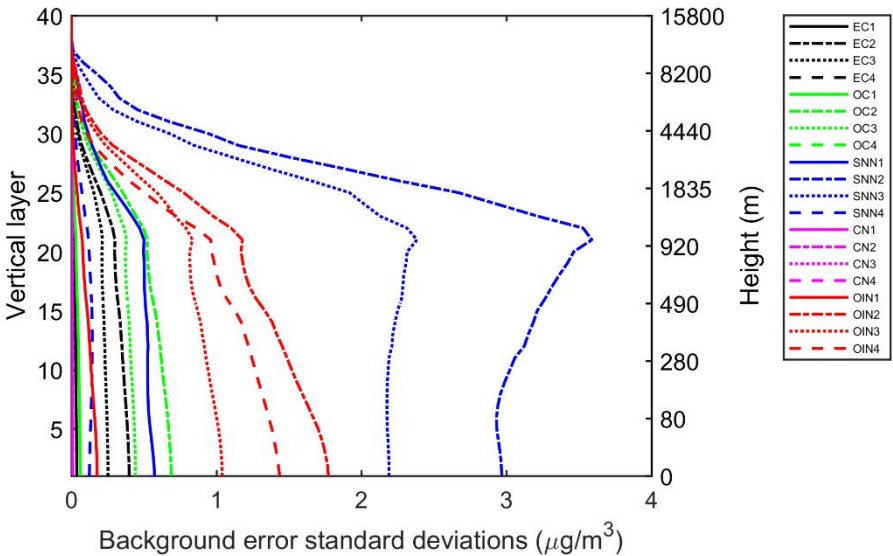

**Figure 4. Vertical profiles of background error STDs in mass concentration for aerosol control variables EC1, EC2, EC3, EC4, OC1, OC2, OC3, OC4, SNN1, SNN2, SNN3, SNN4, CN1, CN2, CN3, CN4, OIN1, OIN2, OIN3, OIN4 in data assimilation process, which were calculated using WRF-Chem D02 forecasts for one month, i.e., November 2018.**









**Figure 5. Vertical auto-correlations of background errors for aerosol state variables within the third size bin, that is, EC3, OC3 SSN3, CN3, OIN3. These statistics are directly estimated by the NMC method using WRF-Chem D02 outputs. Both axes are logarithmic and the contour interval is 0.1.**






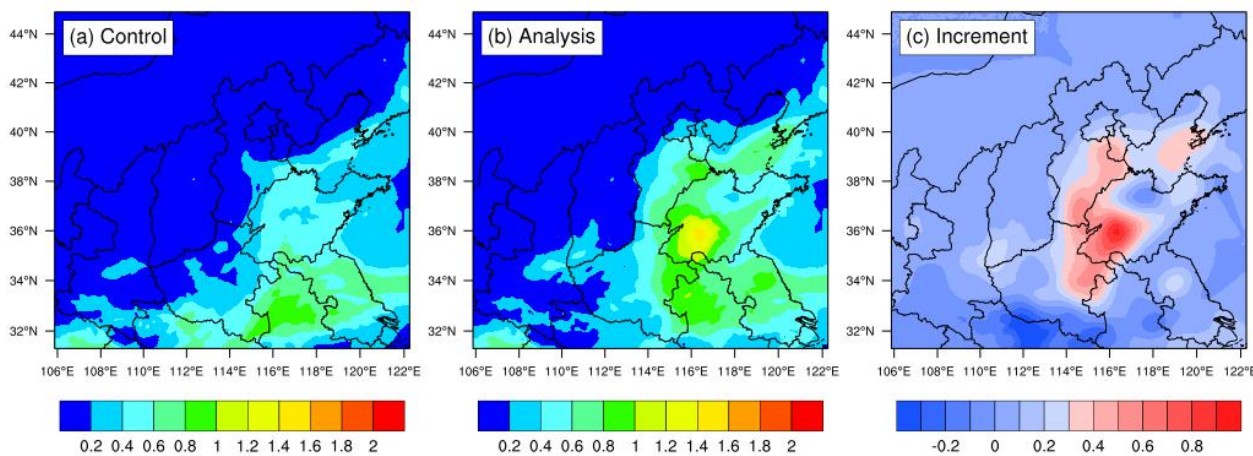

**Figure 6. Spatial distributions of simulated AOTs in the background field (a) and analysis (b), and the increments (c), which are differences between the analysis and the background field. For illustration, distributions in D02 at a model initialization of 0300 UTC on 25 November 2018 are given, which are similar to other results during the experiment period (i.e., from 23 to 29 November 2018).**







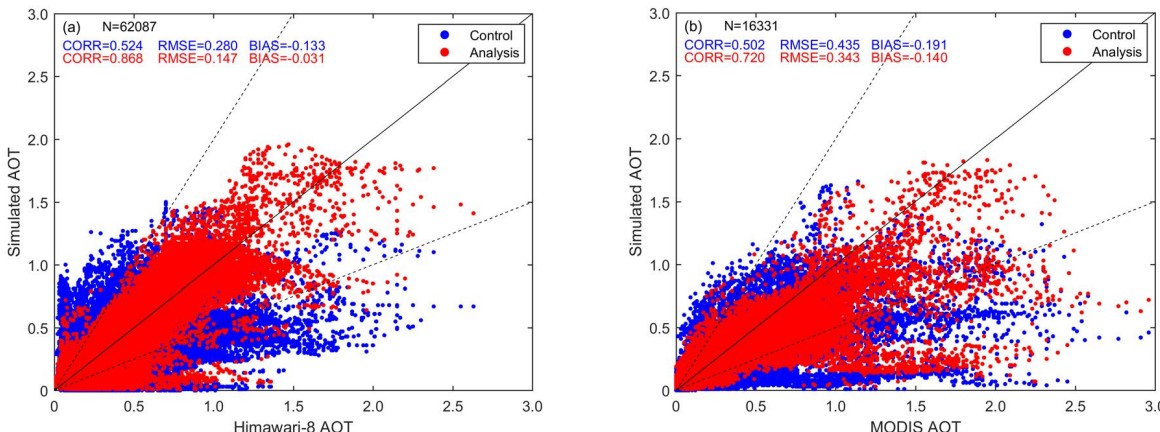

**Figure 7. Scatter plots of the simulated AOTs collocated in D02 versus (a) the observed Himawari-8 AOTs and (b) Terra MODIS AOTs. These data were a set of all initializations from 23 to 29 November 2018, blue points are the control experiment while red points are the assimilation experiment. The solid line is the 1:1 line where simulated values are equal to observed values, and the dashed lines correspond to 1:2 and 2:1.**







**Figure 8. Time series of the simulated AOTs collocated in D02 and AERONET AOT observations at (a) Beijing, (b) Beijing-CAMS, (c) Beijing_PKU, (d) Beijing_RADI, (e) XiangHe, and (f) XuZhou-CUMT during the whole forecasting period. Both simulated AOTs and observed AOTs are at 500 nm. The brown line is the control experiment while the light blue line is the assimilation experiment and the AERONET observations are represented by black dots, which are only available under clear-sky conditions.**




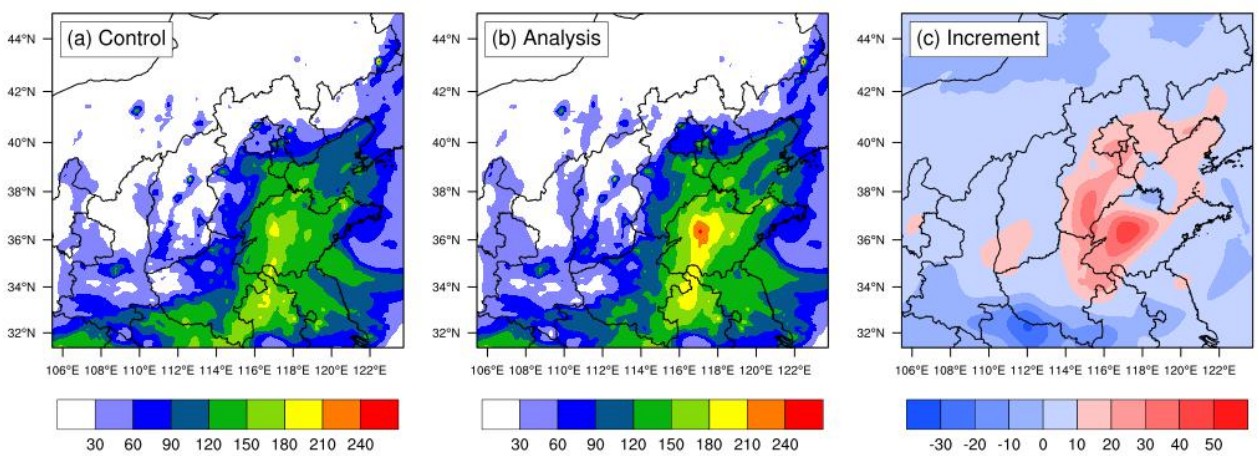

**Figure 9. Spatial distribution of surface PM$_{2.5}$ concentrations simulated at an initialization of 0300 UTC on 25 November 2018 in (a) the control experiment and (b) the assimilation experiment as well as (c) the increment that is the difference between (b) and (a) These quantities are in unit of μg m$^{-3}$ and collocated in D02.**







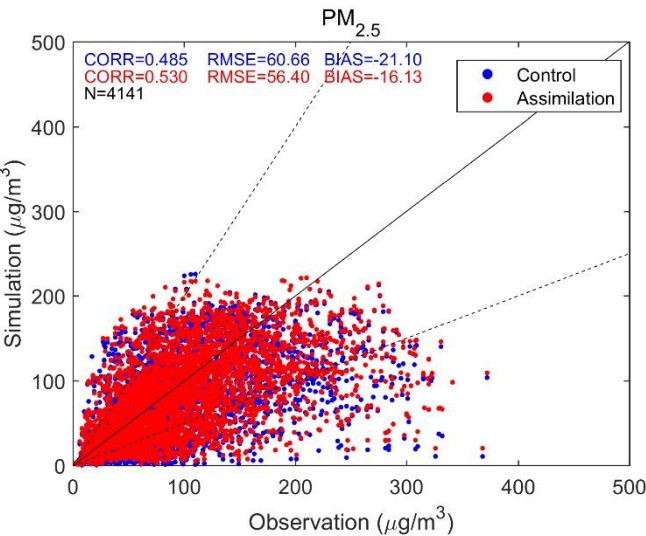

**Figure 10. Scatter plots of the simulated PM$_{2.5}$ concentrations in the control experiment and corresponding analyses in the assimilation experiment versus the observations. Like Figure 7, these data are also collocated in D02 and a set of all initializations. Blue points stand for the control experiment while red points stand for the assimilation experiment.**







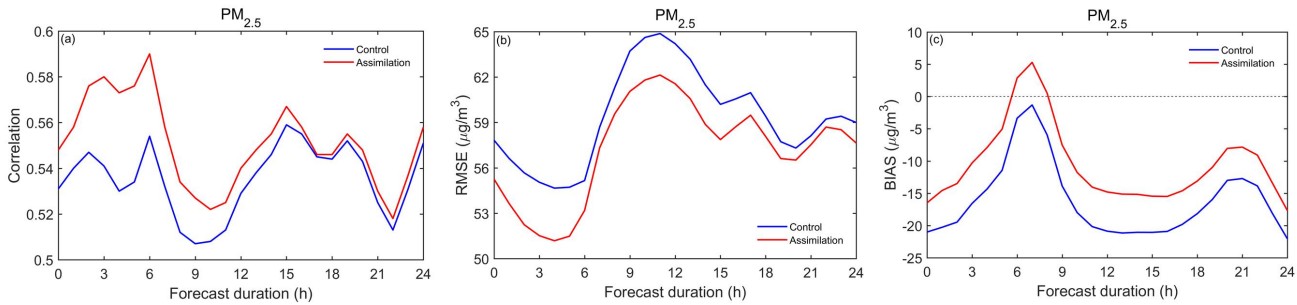

**Figure 11. Statistical metrics (a) CORR, (b) RMSE, and (c) BIAS for surface PM$_{2.5}$ forecast performances in D02 regarding the forecast range, which are computed as an average over 7 single experiments. Likewise, the blue line is the control experiment and the red one is the assimilation experiment.**





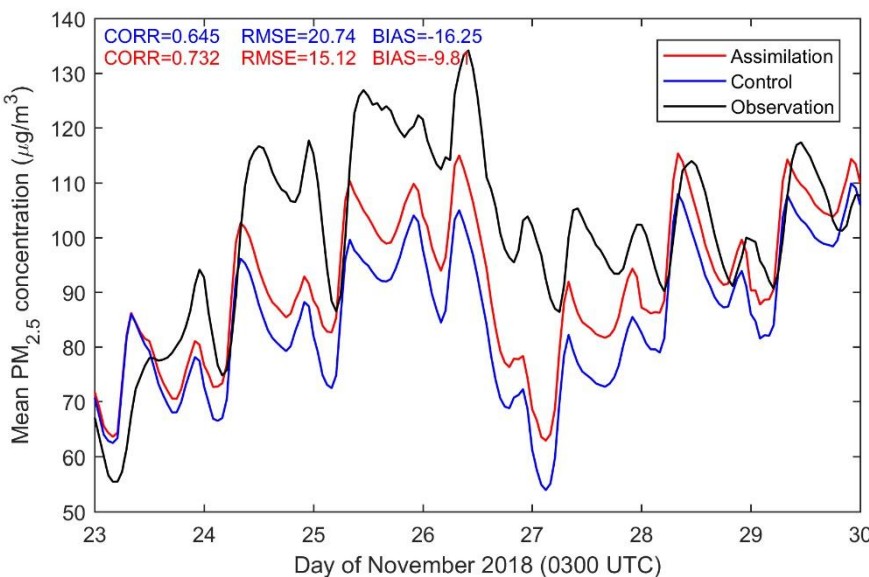

**Figure 12. Time series of surface PM₂.₅ simulated by the control experiment (blue) and the assimilation experiment (red) as well as corresponding observations (black), as averages over 683 stations in D02. The simulations are representative of hourly 0-23 h forecasts in D02 from 0300 UTC every day during the whole forecasting period using WRF-Chem.**
