# Peer review of "A three-dimensional variational data assimilation system for aerosol optical properties based on WRF-Chem v4.0: design, development, and application of assimilating Himawari-8 aerosol observations"

_Geoscientific Model Development, 2021_

## Referee Comment (RC1)

**General comments**

Dear editor and authors,

The manuscript presents an interesting application of aerosol data assimilation for a high pollution event in the eastern China for the period 23 to 29 November 2018. Aerosol Optical Depth (AOD) from the Himarawari-8 satellite is assimilated in an effort to improve AOD and surface particulate matter smaller than 2.5µm (PM2.5). Assimilation experiment is evaluated with independent observations (MODIS, AERONET and China's nationwide monitoring network system) and shows improvement in comparison to Control experiment, either for AOD or PM2.5.

Overall the manuscript is well written, describes most of the aspects of the system in detail and the result section is clear and well presented. However the authors are not discussing adequately the role of observation representation error, the possible misrepresentation of aerosol state when assimilating just AOD while adjusting 20 aerosol state vectors, as well as the impact of the temporal assimilation cycle (1 day). I briefly discuss this three points below. Further I have included some specific comments that can improve quality and readability of the manuscript as well as some technical corrections.

- How can AOD distinguish and constrain 20 different aerosol state variables? What is the impact of using only AOD? There is no mention of other studies that assimilate more information than just AOD (e.g. AOD in other wavelengths or Angstrom Exponent, Absorption Aerosol Optical Depth or Single Scattering Albedo as well as direct radiances assimilation). Although the authors acknowledge the need for combine assimilation of various optical properties in their closing statement in conclusions (L746-751), many recent studies that are related to that are not mentioned. To name a few ones: (Chen et al., 2019; Escribano et al., 2017; Tsikerdekis et al., 2021)

- The spatial aggregation of observations that the authors describe (aggregating observations in the spatial resolution of the model) is indeed often used in data assimilation studies. Although was there any consideration regarding the representation error of this aggregated observations? For example, was the observational error inflated by X amount because you were not using the original resolution of Himawari-8? (Lines 437-442)

- As a geostationary satellite, Himawari-8 is known for its high temporal frequency. Since the data assimilation cycle is in daily frequency (updating analysis once a day), are you fully exploiting this satellite capabilities or rather its strong point? I realize that the daily assimilation step was chosen for practical reasons (computational speed), nevertheless I would expect some discussion about it. Further related to this topic, I did not find any discussion related to temporal collocation of observation in the data assimilation system.

**Specific Comments**

L60: Missing references.

L65-67: Refence, name and accessibility (or the lack of) for this dataset should be provided.

L73: Probably mean "remote sensing optical properties can cover a much larger domain". Because just optical properties can be retrieved also from AERONET stations.

L189-192: In principle PM2.5 can be estimate from the modes that the MADE scheme uses, assuming you know the median and the standard deviation of the distribution for each mode. In that case MADE would be superior to MOSAIC since it will also include mixing of different species within each mode. So I would suggest to emphasize only the numerical efficiency of MOSAIC against MADE. Further, indicating how much faster it is could really promote that argument and it could be easily estimated with two forward simulations, one with MADE one with MOSAIC (no DA required).

L211-213: Authors could mention here that the vertical axis is on hybrid sigma-pressure levels, if that is the case.

L237-238: It would be really helpful to briefly mention here how Yumimoto et al. (2016) estimated this error for Himwari-8 AOD and what this error actually describes (e.g. instrument error, retrieval error, representation error) ?

L491-493: It would be interesting to compare the D02 and D01 estimated background error standard deviation. It would show how important is the model horizontal resolution for this metric. If possible an additional plot for the D01 over the domain of D02.

L562: I would strongly recommend to replace "improvements" with "changes" in that sentence or rephrase. Figure 6 shows the differences of the Analysis – Control. It is not an evaluation with observations (assimilated or independent) where we can truly determine if there was an improvement by the data assimilation.

L585-587: It would be beneficial to provide how much this difference in AOD wavelength (500nm and 550nm) is affecting your evaluation. Maybe you can use Angstrom Exponent from AERONET to determine that and provide a number? Usually AOD at higher wavelength (550nm) is smaller than AOD at lower wavelength (500nm). Which means that the bias would be even more negative if you were comparing MODIS and Model at the same wavelength at Figure 7b. I think it is worth discussing in the manuscript (L595+) although it may enhance the negative bias you get for both Control and Analysis.

L604-606: AERONET sites at Figure 1b are hardly visible (probably because 4 of them are in the Beijing area). It would be visually better to enlarge them a bit.

L664-669: Good point, spatial availability of AOD in contrast to PM2.5 can play a role. I would also add that AOD is an atmospheric column measurement while PM2.5 is a surface measurement. Therefore, if you have an aerosol plume which is not close to the surface AOD

can be increased by increasing the aerosol concentration of that plume while PM2.5 can remain almost unaffected by that change.

**Technical Corrections**

L140: "3DAVR" to "3DVAR"
L173: "back carbon" to "black carbon"
L203: "/MADE/" is some kind of typo?
L291: "black carton, organic carton" to "black carbon, organic carbon"
L609: Something is missing in the sentence. Probably "used to" to "used them to"
L1185: Figure 11: Do you mean "average over 7 analysis steps" instead of "average over 7 single experiments"?

**References**

Chen, C., Dubovik, O., Henze, D. K., Chin, M., Lapyonok, T., Schuster, G. L., Ducos, F., Fuertes, D., Litvinov, P., Li, L., Lopatin, A., Hu, Q. and Torres, B.: Constraining global aerosol emissions using POLDER/PARASOL satellite remote sensing observations, Atmos. Chem. Phys., 19(23), 14585–14606, doi:10.5194/acp-19-14585-2019, 2019.

Escribano, J., Boucher, O., Chevallier, F. and Huneeus, N.: Impact of the choice of the satellite aerosol optical depth product in a sub-regional dust emission inversion, Atmos. Chem. Phys., 17(11), 7111–7126, doi:10.5194/acp-17-7111-2017, 2017.

Tsikerdekis, A., Schutgens, N. A. J. and Hasekamp, O. P.: Assimilating aerosol optical properties related to size and absorption from POLDER/PARASOL with an ensemble data assimilation system, Atmos. Chem. Phys., 21(4), 2637–2674, doi:10.5194/acp-21-2637-2021, 2021.

---

## Referee Comment (RC2)

Title: A three-dimensional variational data assimilation system for aerosol optical properties based on WRF-Chem: design, development, and application of assimilating Himawari-8 aerosol observations
Author(s): Daichun Wang et al.
MS No.: gmd-2021-215
MS type: Development and technical paper

**General Comments:**

The authors developed a new capability of assimilating aerosol optical properties, including AOD, aerosol mass concentration, and aerosol backscatter data, using a 3DVAR method. The analysis system was developed to work with the MOSAIC chemistry option inside the WRF-Chem model. They used one severe air pollution episode that occurred in North China during November 23-29, 2018, to demonstrate the new development. Only the AOD data from Himawari-8 were assimilated. Two numerical experiments were conducted, and their forecasts were initialized by different aerosol data. One used aerosol analysis after 24-h data cycling of AOD assimilation (called Assimilation) and the other one used aerosol data from the previous 24-h forecast (called Control). Results are compared with different observations, including AOD from Himawari-8, Terra MODIS, and AERONET and surface PM2.5 concentration, and are statistically evaluated using the correlation coefficient, RMSE, and mean bias. The assimilation of AOD can improve aerosol forecasts for about 24 h.

The new development of assimilating aerosol optical properties should be encouraged. In particular, the development of assimilating more optical properties, such as backscatter data, can be useful, though no data assimilation experiments have been tested, except for AOD. The authors provide a lot of details on their analysis system development, which is great but could be also too tedious, depending on the background of readers. Nevertheless, the documentation of the system development will certainly be appreciated by some readers. I have a few comments to further improve the manuscript.

**Major comments:**

- Using a constant observational error covariance of 0.06 seems not very convincing. For AOD of 1.8, the error is only 3.3%. Is this realistic? The observational error plays an important role in the DA analysis. Some justification for using this value is needed.

- More detailed information in numerical experiment design is needed. Is AOD DA performed every hour whenever AOD data are available? Does the forecast last for 24 h only? For each 24-h DA cycle, are the meteorological data in the first guess from FNL or from data at the end of the previous cycle? Similarly, for each forecast starting at 0300 UCT, while aerosols are taken from the analysis after a 24-h DA cycle for the Analysis run and from the previous 24-h forecast for the Control run, are meteorological conditions taken from FNL?

- The development of assimilating optical properties was built on the framework of Li et al. (2013). The authors should discuss major differences between the two analysis systems and major differences in the conclusions of the two studies.

- The improvement of aerosol forecasts only lasts for 24 hours in this study. Although Li et al. (2013) also showed a similar result, this seems a little bit short in terms of forecast length. Some studies have shown the benefit of assimilating AOD data in longer aerosol forecasts (48 h), such as Benedetti, et al. 2019 and Choi et al. 2020. Could it be due to, for example, no assimilation of meteorological data, the quality of AOD data, the assimilation method, the study location, etc.? The authors should compare their results with others' or make some comments about this issue (24 h versus 48 h).

*Benedetti, A., Di Giuseppe, F., Jones, L., Peuch, V.-H., Rémy, S., and Zhang, X.: The value of satellite observations in the analysis and short-range prediction of Asian dust, Atmos. Chem. Phys., 19, 987–998, https://doi.org/10.5194/acp-19-987-2019, 2019.*

*Choi, Y., Chen, S.-H., Huang, C.-C.,Earl, K., Chen, C.-Y., Schwartz, C. S., &Matsui, T. (2020). Evaluating theimpact of assimilating aerosol opticaldepth observations on dust forecastsover North Africa and the East Atlanticusing diff erent data assimilation meth-ods. Journal of Advances in ModelingEarth Systems, 12, e2019MS001890*

- **Minor comments:**
  1. Line 65. "… monitoring, for instance, China has…" should be ""… monitoring. For instance, China has…"
  2. Line 74. "… detailed aerosol profiles (Kaufman et al., 2002), …" Kaufman et al., 2002 used AOT and aerosol index for their study. Both are vertically integrated data and thus do not provide vertical profile information.
  3. Line 98. What does the "control variable scheme" mean? DA methods usually need control variables. Do you mean "...PM10, which is used as a control variable?"
  4. Lines 120-122. I believe that ECMWF uses a 4DVAR method to assimilate AOD and it is an online approach. Check out Benedetti et al. 2019 paper listed above.
  5. Lines 236-237. "…observation errors associated with AOD retrievals are determined by measuring instruments…" It is probably more than just the instrument itself, but also the retrieval algorithm and surface emissivity, to name a few.
  6. Line 261. Define BEGS.
  7. Lines 440 and 442. The data reduction used in this study is not a thinning procedure but a superobbing procedure.
  8. Line 457. Add "AOT" in front of assimilation.
  9. Line 569. "… with negative increments marked in blue." Improve the color shading in Figure 6c. Make warm and cold colors for positive and negative values, respectively. The current plot mixes red and blue colors for positive values, while it uses blue shading for negative values. This is confusing. A similar problem is seen in Figure 9c.

10. Line 594. "… BIAS increase…" This statement sounds like that the assimilation of AOD data makes the result worse, but it is not true. Need to rewrite this. The same for line 663.
11. Try to use words consistently throughout the paper, such as "cost function" versus "objective function", "AOD" versus "AOT", "Control" versus "control" experiment, and "Assimilation" versus "assimilation" experiment.

---

## Referee Comment (RC3)

**Review of "A three-dimensional variational data assimilation system for aerosol optical properties based on WRF-Chem: design, development, and application of assimilating Himawari-8 aerosol observations" by Wang et al. (gmd-2021-215)**

**General comments**

This study proposes a new development in assimilation of aerosol optical properties based on the foundation of three-dimensional data assimilation (3DVAR) scheme in the GSI and coupled with framework of MOSAIC in WRF-Chem. A heavily polluted event of one-week period occurred in northern China is selected for the demonstration. Referenced by several observational datasets, the results indicate that assimilation of Himawari-8 AOT retrieval can reduce the negative bias in modeling both AOT and PM2.5 over the region in comparison to the simulation without any aerosol DA.

In general, this manuscript is well-organized and quite readable. I enjoy reading the technical descriptions which provides sufficient information even for entry-level reader. Nevertheless, I realize the discussions in the sources of uncertainties in aerosol modeling, assimilation frequency of satellite data, and analyzed vertical distribution of aerosol properties are not included in current form and may need to be addressed to further enrich the contents. Thus, three major comments are made accordingly and provided below. Specific comments and suggestions are also given in the followed list.

1. The model resolution, meteorological conditions, and emission data could be other important sources of uncertainty in the air pollution modeling and in fact some of them can be identified in the diagram you show in Fig. 2. However, they are not discussed in the manuscript. Would you be able to quantify these uncertainties in relation to the impact of aerosol field initialization (DA) based on the design of model experiment? For instance, you may consider conducting additional experiment which assimilate meteorological states and aerosol to explore their relative impacts on the subsequent forecast.

2. The under-utilization of Himawari-8 AOT product (hourly data) in the context of assimilation frequency (24 h) seems to be obvious. I imagine a strategy with more frequent assimilations could be a unique point to make in this research as the geostationary satellite product used here has such a high temporal resolution. Nevertheless, the relevant discussion is not covered in the manuscript. I would suggest adding more content to address this comment.

3. Despite the vertical profiles of background error STDs and auto-correlations are given, the analyzed increments of each aerosol state variables are not seen anywhere in the document. Since the AOD is obtained through the integration of aerosol properties in the atmospheric column, it would be useful to show analyzed results in terms of their vertical

distributions and further discuss how would that contribute to the uncertainty of simulation.

**Specific comments**

L32: It is mentioned here that the developed DA system is able to assimilate lidar-based aerosol profiles. However, I did not find any relevant description with respect to the treatment in the followed sections. Would you clarify this?

L237-240: Have you conducted any experiment to test how sensitive this constant error is?

L260: Can you give an example of the minimization process, such as reduction of cost function in function of iteration numbers?

L288-289: Please include references to supplement statement here.

L291: Should be black car"b"on and organic car"b"on

L369: Would this introduce any inconsistency between nonlinear model and TL? Also, I am curious how did you deal with if statements in the code if there's any.

L389: Since this manuscript documents the development of a DA package, it is of necessity to show the result of TL/AD test. For example, it is common to show the plot of gradient check with respect to various orders of perturbation.

L418: Please cite this reanalysis product and provide the link of the data source.

L422: The assimilation cycle time (24 hours) seems to be coarse in relation to data availability. Please discuss how it is designed and clarify if there's any limitation on the data coverage or quality, etc.

L424-426: The statement here is contradictory to the design of assimilation cycles. Please explain.

L441: I am not sure this is the best treatment as it could further smooth out the observed data. Please address.

L443 and L463: Fig. 3b is mentioned earlier than Fig. 3a. I would suggest swapping them for the fluency of reading.

L492: It looks like the similar DA procedure is also carried out over the D01 but at least with different treatment in data thinning. Have you done any experiment without assimilation in D01? If true, what was the impact of additional DA in D01.

L532: Is it possible to estimate the correlation length with the observational data or alternatively the analysis after assimilation?

L577-579: Sentences such as these in the manuscript could be trimmed to shorten the length.

L587: Please elaborate more on this. Would the uncertainty mostly be on the magnitude or something else?

L606: The red triangles in Fig. 1b are hardly distinguished from one another as they are basically overlapped with each other. Please try to make them more visible. Add another zoomed-in map may help achieve that.

L610: What is the temporal resolution of AERONET observations? From the time series plot of Fig. 8, it looks like the data is mostly only available around 00 UTC of each day

L615-616: Any explanation why model has worse skill at XuZhou-CUMT? It seems the event on Nov. 25 is more severe than Nov. 26 at this site and not captured as well.

L617-618: Any guess on this? Have you looked at the meteorological conditions on these days? Could it be associated with the intensity of wind speed?

L622: It would be easier for reader to understand if the data distribution map of Nov. 26 is also provided. Along the same line, I would suggest adding information of available data amount in Fig. 8 to address this.

L643: You may remove "between analyses and the background field" since increment has been defined in the earlier paragraph.

L644-645: The of color bar scales in Fig. 3a and Fig. 9 are not consistent, which makes it hard to compare them visually. Please consider modify them.

L645: Need to mark where Tianjin is in the map, otherwise one may not know which location you talked about.

L644: Panels in Fig. 9 are not sufficient to conclude the underestimation in control experiment as no observation is provided.

---

## Author Comment (AC2)

**Responses to the comments of Reviewer #1:**

We are truly grateful to yours' positive comments and thoughtful suggestions. Those comments are all valuable and very helpful for revising and improving our paper, as well as the important guiding significance to our researches. Based on these comments and suggestions, we have studied comments carefully and have made correction which we hope meet with approval. All changes made to the text are marked in red color. Below you will find our point-by-point responses to the reviewers' comments/questions:

**General comments:**

*1. How can AOD distinguish and constrain 20 different aerosol state variables? What is the impact of using only AOD? There is no mention of other studies that assimilate more information than just AOD (e.g. AOD in other wavelengths or Angstrom Exponent, Absorption Aerosol Optical Depth or Single Scattering Albedo as well as direct radiances assimilation). Although the authors acknowledge the need for combine assimilation of various optical properties in their closing statement in conclusions (L746-751), many recent studies that are related to that are not mentioned. To name a few ones: (Chen et al., 2019; Escribano et al., 2017; Tsikerdekiset al., 2021)*

**Response:**

Thank you very much for your questions and suggestions. First, the forward observation operator links aerosol optical properties (including AOD, extinction coefficient, backscattering coefficient, and total attenuated backscattering coefficient) with 20 different state variables in the data assimilation system, which means that AOD observations distinguish and constrain 20 different state variables via the forward operator. Designing and establishing the observation operator is crucial to directly assimilate optical properties in case that control or state variables are mass concentrations instead of optical properties. Fortunately, we can reduce the aerosol Optical Module within WRF-Chem to establish the forward operator, which is based on the Mie-scatter theory. Different aerosol species described by 20 aerosol state variables here make greatly different contributions to AOD, even for the same species, particles within different size bins make different contributions. The operator can quantify these contributions. Specifically, AOD can constrain particle size and number, and then adjust individual species mass concentrations denoted by 20 different aerosol state variables. Second. Only AOD observation was chosen to test the developed assimilation system, its impact may be insufficient for significantly improving aerosol forecasts. It is noted that the developed assimilation system can assimilate extinction and backscattering profiles, AOD, and attenuated backscattering at different wavelengths because the wavelength is designed as a variable parameter in the assimilation system when establishing the observation operator, but it can not assimilate other optical properties such as Angstrom Exponent, Absorption Aerosol Optical Depth or Single Scattering Albedo as well as direct radiances (Assimilating aerosol direct radiance is very challenging because it is affected by many factors). Nevertheless, we will attempt to combine assimilate more aerosol optical properties to constrain model variable more accurately in the near future work. Finally, some recent studies related to combined assimilation of various optical properties have been added in the revised version as "With the increase in aerosol observations, the simultaneous assimilation of aerosol observations from various platforms has become a trend, in particular combined assimilation of various optical properties has made great progress in recent year (Escribano et al., 2017; Chen et al., 2019; Tsikerdekiset al., 2021)."

*2. The spatial aggregation of observations that the authors describe (aggregating*

*observations in the spatial resolution of the model) is indeed often used in data*

*assimilation studies. Although was there any consideration regarding the*

*representation error of this aggregated observations? For example, was the*

*observational error inflated by X amount because you were not using the original*

*resolution of Himawari-8? (Lines 437-442)*

**Response:**

We really appreciate your valuable suggestion. We aggregated AOT observations in the spatial resolution of the model, which is also employed by other researchers (Yumimoto et al., 2016; Dai et al., 2019; Ha et al., 2020). The observation error plays an important role in assimilation process. In general, the observation error depends on measurement error and representation error, however, it is very difficult to accurately determine the representation error because the released AOT product gives the retrieval uncertainty rather than representation error, what is more, the retrieval uncertainty is just a reference range. Consequently, the observation error here can only be roughly determined based on experience or tuning parameter. Aggregating

AOT observations by averaging them in one grid cell can not inflate observation error, conversely, this approach can smooth out much noise to improve the quality. At least, the assimilation practice has demonstrated that assimilating aggregated AOT

observations is better than original observations.

*3. As a geostationary satellite, Himawari-8 is known for its high temporal frequency.*

*Since the data assimilation cycle is in daily frequency (updating analysis once a*

*day), are you fully exploiting this satellite capabilities or rather its strong point? I*

*realize that the daily assimilation step was chosen for practical reasons*

*(computational speed), nevertheless I would expect some discussion about it.*

*Further related to this topic, I did not find any discussion related to temporal*

*collocation of observation in the data assimilation system.*

**Response:**

We really appreciate your suggestion. Himawari-8 level 3 AOT_Merged, an improved hourly product, which is derived from level 2 AOT retrievals at a 10 min interval, was employed to conduct assimilation experiments. A daily assimilation frequency seems to be an underutilization of Himawari-8 observations in comparison to its high temporal frequency. Since AOT observations are retrieved at the visible and infrared bands, observations between 03 and 08 UTC in the daytime are available for

China. In fact, AOT observations are noticeably noisy, which will have a greatly negative impact on assimilation results. What is more, observations at afternoon are much noisier than those in the morning. For example, surface $PM_{2.5}$ concentration and original (not thinned) Himawari-8 AOT observations at 0300 UTC and 0600 UTC are plotted in Fig. 1 and Fig. 2, respectively. Overall, surface $PM_{2.5}$ mass concentrations change little even with a small decrease at some areas from 0300 to 0600 UTC (Fig.

1b, Fig. 2b) while there is a remarkably increase in AOTs during the same period (Fig.

1a, Fig. 2a). In terms of $PM_{2.5}$, the noticeably increase in AOT observations should not be considered as normal changes of aerosol but much noises. As a result, more frequent assimilation of AOT observations like this will certainly result in a dramatic overestimation of $PM_{2.5}$ mass concentrations. In terms of evaluation with $PM_{2.5}$ mass concentration observations, AOT observations at 0300 UTC without no temporal collocation were only assimilated in this study to test the developed assimilation system. As known, data assimilation serves only as a mathematical approach on how to introduce observations into the model, and then improves model initializations and forecasts. Assimilation results are largely determined by the quality of observational data, as for how to deal with those with high noise and improve the quality, more researches are needed in the future. Moreover, the advanced DA system such as

4DVAR will be developed in the future that can assimilate observational data from a time window.

[Figure]

**Figure 1. Observations of the original (not thinned) Himawari-8 AOTs (a) and surface PM$_{2.5}$ mass concentration (b) in D02 at 0300 UTC on 25 November 2018.**

[Figure]

**Figure 2. Same as Fig.1, but at 0600 UTC on 25 November 2018.**

**Specific Comments:**

*4. L60: Missing references.*

**Response:**

We really appreciated the suggestion and followed it. Three references have been added here (L61).

Menon, S., Hansen, j., Nazarenko, L., and Luo, Y.: Climate Effects of Black

Carbon Aerosols in China and India, Science, 297, 2250–2253.

https://doi.org/10.1126/science.1075159, 2002.

Gao, M., Guttikunda, S. K., Carmichael, G. R., Wang, Y., Liu, Z., Stanier, C. O.,

Saide, P. E., and Yu, M.: Health impacts and economic losses assessment of the 2013

severe haze event in Beijing area, Sci. Total. Environ., 511, 553−561, https://doi.org/10.1016/j.scitotenv.2015.01.005, 2015.

Qian, Y., Gong, D., Fan, J., Leung, L.R., Bennartz, R., Chen, D., and Wang, W.:

Heavy pollution suppresses light rain in China: Observations and modeling, J.

Geophys. Res., 114, D00K02, https://doi.org/10.1029/2008JD011575, 2009.

*5. L65-67: Reference, name and accessibility (or the lack of) for this dataset should*

*be provided.*

**Response:**

We really appreciate your valuable suggestion. This dataset is provided by China

National Environmental Monitoring Centre (CNEMC) but has no official name. This sentence has been revised as "For instance, China National Environmental Monitoring

Centre (CNEMC, http://www.cnemc.cn/en/) has established a nationwide monitoring network consisting of more than 1500 stations since 2013 to provide near-time data of pollutants, including $PM_{2.5}$, $PM_{10}$, $SO_2$, $NO_2$, CO, and $O_3$."(L66-67)

*6. L73: Probably mean "remote sensing optical properties can cover a much larger*

*domain". Because just optical properties can be retrieved also from AERONET*

*stations.*

**Response:**

Thank you so much for your valuable suggestion. The sentence has been revised as "Remote sensing optical properties can cover a much larger domain (Kaufman et al., 2002) and provide detailed aerosol profiles (Young and Vaughan, 2009)" (L75-76), at the same time, this reference has been added in the revised manuscript ("Young, S. A. and Vaughan, M. A.: The retrieval of profiles of particulate extinction from Cloud-Aerosol Lidar Infrared Pathfinder Satellite Observations (CALIPSO) data: Algorithm description, J. Atmos. Ocean. Tech., 26, 1105–1119, https://doi.org/10.1175/2008JTECHA1221.1, 2009.")

*7. L189-192: In principle PM$_{2.5}$ can be estimate from the modes that the MADE scheme uses, assuming you know the median and the standard deviation of the distribution for each mode. In that case MADE would be superior to MOSAIC since it will also include mixing of different species within each mode. So I would suggest to emphasize only the numerical efficiency of MOSAIC against MADE. Further, indicating how much faster it is could really promote that argument and it could be easily estimated with two forward simulations, one with MADE one with MOSAIC (no DA required).*

**Response:**

Thank you so much for your valuable suggestion. We agree well with you. Due to its simplicity and high numerical efficiency, the MOSAIC scheme has been chosen to develop the data assimilation system. Consequently, it seems to unnecessary to discuss how much faster is MOSAIC against MADE for aerosol simulations in the context of testing the assimilation system.

*8. L211-213: Authors could mention here that the vertical axis is on hybrid sigma-pressure levels, if that is the case.*

**Response:**

We followed this suggestion and this sentence has been revised as "To ensure a detailed simulation of aerosol vertical distributions, 40 vertical layers were modelled in the simulation, and it is worth mentioning that the vertical axis is on hybrid sigma-pressure levels with a resolution decreasing with height. The lowest layer is at the surface, whereas the top reaches 50 hPa". (L214-215)

*9. L237-238: It would be really helpful to briefly mention here how Yumimoto et al.*

*(2016) estimated this error for Himwari-8 AOD and what this error actually*

*describes (e.g. instrument error, retrieval error, representation error) ?*

**Response:**

Thank you so much for your valuable suggestion. Yumimoto et al. (2016)

estimated observation errors to be the retrieval uncertainty attached to the Himawari-8

AOT data plus a standard deviation calculated as the representative error in the regridding (Zhang et al., 2008, see below). The retrieval uncertainty ranged from

0.0001 to 1.04 with average of 0.013 and has larger values in the land relative to over the ocean.

The observation error plays an important role in assimilation process, however, no relevant theoretical basis has been found so far. The observation error depends on measurement error and representation error (Elbern and Schmidt, 2001; Schwartz et al.,

2012; Jiang et al., 2013), nevertheless, how to determine the observation error is also a matter of assimilation practice. Because the observation error determines the weight of observation across the analysis, that is, the smaller the observation error, the greater the absolute value of the assimilation incremental field are, and the closer the assimilation analysis field are to the observation field deviating from the background field. In other words, no matter how large the observation error is, as long as the observation operator is correct, the assimilation analysis field will always fall between the background field and the observation field and has a positive assimilation effect, even though not the best.

In this study, AOT observation error was set to be a simple value which is rational only to test the developed assimilation system.

Zhang, J., Reid, J. S., Westphal, D. L., Baker, N. L., and Hyer, E. J.: A system for operational aerosol optical depth data assimilation over global oceans, J. Geophys. Res.,

113, D10208, https://doi.org/10.1029/2007JD009065, 2008.

*10. L491-493: It would be interesting to compare the D02 and D01 estimated*

*background error standard deviation. It would show how important is the model*

*horizontal resolution for this metric. If possible an additional plot for the D01*

*over the domain of D02.*

**Response:**

We really appreciated the suggestion. Because both D01 and D01 outputs were assimilated using AOT observations in this study, background error covariance including standard derivation and correlation was estimated in D01 and D02, respectively. Only the estimated background error standard deviation in D02 was shown in manuscript, as shown in Fig .3b here, the D01 estimated background error standard deviation looks actually like D02, as shown in Fig. 3a. Obviously, the D02

estimated background error standard deviation is nearly twice than D01 estimated ones, whereas the D01 model horizontal resolution is 27km and D02 is 9km. The background error standard deviation determines the magnitude of analysis increments across aerosol control variables. As these two plots look alike, it seems unnecessary to add the plot for D01.

[Figure]

**Figure 3. Vertical profiles of background error standard deviation in mass concentration for aerosol control variables, (a) is for D01, and (b) is for D02.**

*11. L562: I would strongly recommend to replace "improvements" with "changes" in that sentence or rephrase. Figure 6 shows the differences of the Analysis – Control. It is not an evaluation with observations (assimilated or independent) where we can truly determine if there was an improvement by the data assimilation.*

**Response:**

The word "improvements" has been replaced by "changes" (L571).

*12. L585-587: It would be beneficial to provide how much this difference in AOD wavelength (500nm and 550nm) is affecting your evaluation. Maybe you can use Angstrom Exponent from AERONET to determine that and provide a number? Usually AOD at higher wavelength (550nm) is smaller than AOD at lower wavelength (500nm). Which means that the bias would be even more negative if you were comparing MODIS and Model at the same wavelength at Figure 7b. I think it is worth discussing in the manuscript (L595+) although it may enhance the negative bias you get for both Control and Analysis.*

**Response:**

We really appreciated the suggestion and followed it, AOD simulation was performed at a wavelength of 500 nm, the same as Himawari-8 retrievals, whereas

MODIS AOD is retrieved at 550 nm. Even though this difference in AOD wavelength may affect the evaluation, it is naturally convincing to evaluate AOD simulation directly employing MODIS AOD because the wavelength difference is minor.

There is no doubt that your suggestion will certainly improve the manuscript, and the following information has been added in the revised manuscript (L607-612).

Usually AOD at higher wavelength (550 nm) is smaller than AOD at lower wavelength (500 nm), so the bias would be even more negative if comparing AOD

simulations with MODIS AOD for both Control and Analysis, which is demonstrated by the indicator BIAS in Fig. 7. For instance, BIAS is -0.031 when comparing with

Himawari-8 AOD, while BIAS is -0.140 against MODIS AOD after assimilation.

*13. L604-606: AERONET sites at Figure 1b are hardly visible (probably because 4 of*

*them are in the Beijing area). It would be visually better to enlarge them a bit.*

**Response:**

We really appreciated and followed the suggestion, and have added a zoomed-in map as Fig. 1c for AERONET sites in Beijing area in the revised version, which is also given as Fig. 4 below:

[Figure]

**Figure 4. A zoomed-in map for AERONET sites in Beijing area, including Beijing, Beijing-CAMS, Beijing_PKU, Beijing_RADI, XiangHe.**

*14. L664-669: Good point, spatial availability of AOD in contrast to PM2.5 can play*

*a role. I would also add that AOD is an atmospheric column measurement while*

*PM2.5 is a surface measurement. Therefore, if you have an aerosol plume which is*

*not close to the surface AOD can be increased by increasing the aerosol*

*concentration of that plume while PM2.5 can remain almost unaffected by that*

*change.*

**Response:**

We really appreciated and followed the suggestion, and have added the following descriptions in the revised manuscript (L694-697).

Besides, AOD is an atmospheric column measurement while $PM_{2.5}$ is a surface measurement. Therefore, if you have an aerosol plume which is not close to the surface, AOD can be increased by increasing the aerosol concentration of that plume while $PM_{2.5}$ can remain almost unaffected by that change.

**Technical Corrections:**

*L140: "3DAVR" to "3DVAR"*

**Response:**

Done. (L143)

*L173: "back carbon" to "black carbon"*

**Response:**

Done. (L175-176)

*L203: "/MADE/" is some kind of typo?*

**Response:**

This sentence has been revised as "the Regional Acid Deposition Model, Version

2 (RADM2, Stockwell et al., 1990), the Modal Aerosol Dynamics Model for Europe (MADE, Ackermann et al., 1998)/Second Organic Aerosol Model (SORGAM, Schell et al., 2001) anthropogenic emissions." (L206-207)

*L291: "black carton, organic carton" to "black carbon, organic carbon"*

**Response:**

Done. (L294-295)

*L609: Something is missing in the sentence. Probably "used to" to "used them to"*

**Response:**

Done. (L622)

*L1185: Figure 11: Do you mean "average over 7 analysis steps" instead of "average*

*over 7 single experiments"?*

**Response:**

We really appreciated and followed the suggestion. Two one-week parallel experiments have been performed to evaluate AOD assimilation effects regarding to

24 h regional $PM_{2.5}$ forecasts. For a general assessment, the statistics were averaged over 7 analysis steps. (L1213)

We would like to express our great appreciation to you for the valuable and pertinent comment on our manuscript, which is crucial to improve the quality of our work. We hope that these revisions are satisfactory and that the revised version will be acceptable for publication in Geoscientific Model Development. Thank you very much for your work concerning my paper.

Wish you all the best!

Yours sincerely,

Daichun Wang and Wei You

11/23/2021

---

## Author Comment (AC3)

**Responses to the comments of Reviewer #2:**

We are truly grateful to yours' positive comments and thoughtful suggestions. Those comments are all valuable and very helpful for revising and improving our paper, as well as the important guiding significance to our researches. Based on these comments and suggestions, we have studied comments carefully and have made correction which we hope meet with approval. All changes made to the text are marked in green color. Below you will find our point-by-point responses to the reviewers' comments/questions:

**Major Comments:**

*1.Using a constant observational error covariance of 0.06 seems not very convincing. For AOD of 1.8, the error is only 3.3%. Is this realistic? The observational error plays an important role in the DA analysis. Some justification for using this value is needed.*

**Response:**

We really appreciate your question. The observation error plays an important role in assimilation process, however, no relevant theoretical basis on its construction has been found so far. The observation error depends on measurement error and representation error (Elbern and Schmidt, 2001; Schwartz et al., 2012; Jiang et al., 2013), and is difficult to accurately estimate so that how to determine it is also a matter of assimilation practice. In several studies, the observation error is given by a tuning parameters. Based on the 3DVAR principle, the function of the observation error can be easily analyzed, namely, the observation error determines the weight of observation across the analysis. Given a background field, the smaller observation error produces the greater increments in terms of absolute value to make the analysis closer to observations and away from the background field and vice versa. No matter how large the observation error is, as long as the observation operator is correct, the generated analysis theoretically will fall between the background field and observations, demonstrating a positive assimilation effect, even though not the best.

Consequently, it is inclined to construct the simple observation error to run the assimilation system in practice. It is apparent that using a constant observation error only to test the developed system is rational.

Even though the observation error can be roughly determined based on experience, it is necessary to select a rational value. According to Yumimoto et al.

(2016), the observation error was estimated to be the retrieval uncertainty attached to the Himawari-8 AOT data plus a standard deviation calculated as the representative error in the regridding. The retrieval uncertainty ranged from 0.0001 to 1.04 with average of 0.013 and has larger values in the land relative to over the ocean. Thus it can be seen that using a constant observation error of 0.06 is rational in this study, which is also obtained after several tests. As you mentioned, as for AOD of 1.8, the value seems somewhat irrational, but these high AOD data account for a small proportion during the study period. It should be pointed out that the observation error varies with data values, which also needs some further researches in the future.

*2.More detailed information in numerical experiment design is needed. Is AOD DA*

*performed every hour whenever AOD data are available? Does the forecast last for*

*24 h only? For each 24-h DA cycle, are the meteorological data in the first guess*

*from FNL or from data at the end of the previous cycle? Similarly, for each forecast*

*starting at 0300 UTC, while aerosols are taken from the analysis after a 24-h DA*

*cycle for the Analysis run and from the previous 24-h forecast for the Control run,*

*are meteorological conditions taken from FNL?*

**Response:**

We really appreciate your question. AOD DA is not performed every hour during the period of 0300 UTC to 0800 UTC when the Himawari-8 AOD observations are available for China. AOD observations at 0300 UTC every day from 23 to 29 November 2018 was only assimilated to provide the analysis (L460-461), and the forecast last for 24 h, which means that the assimilation frequency is 24 h. Comparing to its high temporal resolution (an hourly product), the 24-h assimilation frequency seems to be an underutilization of AOD observations. However, the AOD retrievals are found with much noise, which will have a significantly negative impact on assimilation. For example, surface $PM_{2.5}$ concentration and original (not thinned) Himawari-8 AOD observations at 0300 UTC and 0600 UTC are plotted in Fig. 1 and Fig. 2, respectively. Overall, surface $PM_{2.5}$ mass concentrations change little even with a small decrease at some areas from 0300 to 0600 UTC (Fig. 1b, Fig. 2b) while there is a remarkably increase in AODs during the same period (Fig. 1a, Fig. 2a). In terms of $PM_{2.5}$, the noticeably increase in AOD observations should not be considered as normal changes of aerosol but much noise. As a result, more frequent assimilation of AOD observations like this will certainly result in a dramatic overestimation of $PM_{2.5}$ mass concentrations. In terms of evaluation with $PM_{2.5}$ mass concentration observations, AOD observations at 0300 UTC without no temporal collocation were only assimilated in this study to test the developed assimilation system. As known, DA serves only as a mathematical approach on how to introduce observations into the model, and then improves model initial and forecast fields. Assimilation results are largely determined by observational data, as for how to deal with those with much noise and improve the quality, more researches are needed in the future.

Additionally, for each 24-h DA cycle, the meteorological data in the first guess are from FNL, and the meteorological conditions in both the Analysis run and Control run are taken from FNL, meaning that the Analysis run and Control run utilized the same meteorological conditions. It should be noted that meteorological states were not assimilated in this study because the developed DA system has no capacity of assimilating meteorological data, which aims at aerosol DA.

[Figure]

**Figure 1. Observations of the original (not thinned) Himawari-8 AOTs (a) and surface PM$_{2.5}$ mass concentration (b) in D02 at 0300 UTC on 25 November 2018.**

[Figure]

**Figure 2. Same as Fig.1, but at 0600 UTC on 25 November 2018.**

*3.The development of assimilating optical properties was built on the framework of Li*

*et al. (2013). The authors should discuss major differences between the two analysis*

*systems and major differences in the conclusions of the two studies.*

**Response:**

We really appreciate your question. The DA system presented in this manuscript is an upgrade of that developed by Li et al. (2013). Li et al. (2013) developed a

3DVAR aerosol DA system to work with the sectional scheme MOSAIC within

WRF-Chem for the first time. However, it can only assimilate aerosol mass concentrations, including total mass such as $PM_{2.5}$ and $PM_{10}$ and composition mass, without the ability of assimilating aerosol optical properties. In order to develop the

DA system for aerosol optical properties, the basic framework of Li et al. (2013)

including the minimization process as well as the **B**-matrix computation was employed, but new aerosol state variables are designed based on the MOSAIC scheme.

There are a total of 20 state variables in this DA system while there are 5 variables in

Li et al., (2013). More importantly, an optical module consisting of the nonlinear forward operator achieved by simplifying the Optical Module inside the WRF-Chem model and its tangent linear (TL) as well as adjoint (AD) codes has been added in order to directly assimilate optical properties. In the study of Li et al. (2013), $PM_{2.5}$

mass assimilation has a significant improvement for $PM_{2.5}$ initial conditions and its

24-h subsequent forecasts, whereas, this study mainly focus on the validation of the new development with AOD observations and shows that AOD assimilation improves

24-h $PM_{2.5}$ forecasts and model AOD initial simulations.

*4.The improvement of aerosol forecasts only lasts for 24 hours in this study. Although*

*Li et al. (2013) also showed a similar result, this seems a little bit short in terms of*

*forecast length. Some studies have shown the benefit of assimilating AOD data in*

*longer aerosol forecasts (48 h), such as Benedetti, et al. 2019 and Choi et al. 2020.*

*Could it be due to, for example, no assimilation of meteorological data, the quality*

*of AOD data, the assimilation method, the study location, etc.? The authors should*

*compare their results with others' or make some comments about this issue (24 h*

*versus 48 h).*

**Response:**

We really appreciate your suggestion. In short, the benefit of assimilating AOD data can last longer than 48 h in the studies conducted by Benedetti et al. (2019) and Choi et al. (2020), which is in terms of AOD simulations, however, the improvement lasting for 24 h in this study is in terms of $PM_{2.5}$ forecasts. It is obvious that the results can not be comparable. In our study, AOD assimilation significantly improves AOD initializations and simulations, but the improvement for the forecast length is not evaluated. Both Benedetti et al. (2019) and Choi et al. (2019) assimilated MODIS AOD to improve the dust analysis and forecasts. In the study of Choi et al. (2019), only MODIS AOD was employed to evaluate the assimilation benefits, whereas, independent AOD data from two established ground-based networks as well as $PM_{10}$ data from the China Environmental Protection Agency were used in the evaluation in the study of Benedetti et al. (2019). In spite of the better improvement for AOD simulations, the AOD assimilation can only make small adjustments to $PM_{10}$ but is unable to improve the quality of forecast fundamentally.

**Major Comments:**

*5.Line 65. "… monitoring, for instance, China has…" should be ""… monitoring.*
*For instance, China has…"*

**Response:**

We really appreciated and followed your valuable suggestion. This sentence has been revised as "For instance, China National Environmental Monitoring Centre (CNEMC, http://www.cnemc.cn/en/) has established a nationwide monitoring network consisting of more than 1500 stations since 2013 to provide near-time data of pollutants, including $PM_{2.5}$, $PM_{10}$, $SO_2$, $NO_2$, CO, and $O_3$."(L66-67)

*6.Line 74. "... detailed aerosol profiles (Kaufman et al., 2002), ..." Kaufman et al.,*

*2002 used AOT and aerosol index for their study. Both are vertically integrated*

*data and thus do not provide vertical profile information.*

**Response:**

We really appreciated and followed your valuable suggestion. This sentence has been revised as "Remote sensing optical properties can cover a much larger domain (Kaufman et al., 2002) and provide detailed aerosol profiles (Young and Vaughan,

2009)" (L75-76), at the same time, this piece of reference below has been added:

Young, S. A. and Vaughan, M. A.: The retrieval of profiles of particulate extinction from Cloud-Aerosol Lidar Infrared Pathfinder Satellite Observations (CALIPSO) data: Algorithm description, J. Atmos. Ocean. Tech., 26, 1105–1119, https://doi.org/10.1175/2008JTECHA1221.1, 2009.

*7.Line 98. What does the "control variable scheme" mean? DA methods usually need*

*control variables. Do you mean "...PM10, which is used as a control variable?"*

**Response:**

We really appreciate your question. The control variable scheme means how many control variables, one or more, are employed in DA analysis. The early aerosol

DA usually employed a control variable. For example, $PM_{10}$ (mass concentration)

rather than its compositions is directly employed as the control variable so that observation is the control variable self.

*8.Lines 120-122. I believe that ECMWF uses a 4DVAR method to assimilate AOD*

*and it is an online approach. Check out Benedetti et al. 2019 paper listed above.*

**Response:**

We really appreciate your suggestion. ECMWF has incorporated atmospheric composition variables into its 4DVAR meteorological assimilation analysis system.

The aerosol assimilation uses total aerosol mass rather than composition mass as a control variable, and it can only assimilate satellite-derived AODs and work with the global model. The sentence has been revised as "Although the four-dimensional variational (4DVAR) technique has been extensively used in operations (Gauthier at al., 2007; Benedetti et al., 2019), and has also been employed to assimilate atmospheric chemical compositions such as $O_3$, $SO_2$, and CO based on the simple offline chemical transport model (CTM) (Eibern and Schmidt, 1999; Elbern and

Schmidt, 2001), it is greatly challenging to develop a 4DVAR DA system coupled with the sophisticated aerosol model such as MOSAIC because of the high computational cost and complex adjoint model" in the revised manuscript. (L121-127)

*9.Lines 236-237. "…observation errors associated with AOD retrievals are*

*determined by measuring instruments…" It is probably more than just the*

*instrument itself, but also the retrieval algorithm and surface emissivity, to name a*

*few.*

**Response:**

Thank you so much for your correction. This sentence has been revised as "In general, observation errors associated with AOT retrievals are determined by measurement and representation errors (Elbern and Schmidt, 2001; Schwartz et al.,

2012; Jiang et al., 2013)" in the revised manuscript. (L240-241)

*10.Line 261. Define BEGS.*

**Response:**

We are so sorry for the misspelling. It should be written as BFGS. The L-BFGS

algorithm is a limited memory quasi-Newton method for large scale unconstrained optimization, which was developed by four mathematician Broyden, Fletcher,

Goldfarb, and Shanno, BFGS is their initials. The L-BFGS code has been developed at the Optimization Center, a joint venture of Argonne National Laboratory and

Northwestern University (http://users.iems.northwestern.edu/~nocedal/lbfgs.html).

(L264)

*11.Lines 440 and 442. The data reduction used in this study is not a thinning*

*procedure but a superobbing procedure.*

**Response:**

We really appreciate your question. We thinned AOD observations in the spatial resolution of the model, which is also employed by other researchers (Yumimoto et al.,

2016; Dai et al., 2019; Ha et al., 2020). This approach certainly leads to a great data reduction, however, it can smooth out some noise in retrieved data to improve the quality, which is also of great significance for assimilation. At least, the assimilation practice has demonstrated that assimilating thinned AOD observations is better than original observations. More researches on how to thin data with a high spatial resolution are needed in the future.

*12.Line 457. Add "AOT" in front of assimilation.*

**Response:**

Done. (L466)

*13.Line 569. "… with negative increments marked in blue." Improve the color*

*shading in Figure 6c. Make warm and cold colors for positive and negative values,*

*respectively. The current plot mixes red and blue colors for positive values, while it*

*uses blue shading for negative values. This is confusing. A similar problem is seen*

*in Figure 9c.*

**Response:**

Done. The color shadings in both Figure 6c and Figure 9c have been improved in the revised manuscript so that warm and cold colors are for positive and negative values, respectively.

*14.Line 594. "… BIAS increase…" This statement sounds like that the assimilation of*

*AOD data makes the result worse, but it is not true. Need to rewrite this. The same*

*for line 663.*

**Response:**

We followed the suggestion. This statement has been rewritten as "BIAS is reduced by about 77 percent" (L603). The statement in line 663 has also been written as "reducing BIAS by 4.97 ug m$^{-3}$" (L688).

*15.Try to use words consistently throughout the paper, such as "cost function" versus*

*"objective function", "AOD" versus "AOT", "Control" versus "control"*

*experiment, and "Assimilation" versus "assimilation" experiment.*

**Response:**

Done. We used the words "cost function", "AOT", "Control", and "Assimilation"

consistently throughout the paper in the revised form.

We would like to express our great appreciation to you for the valuable and pertinent comment on our manuscript, which is crucial to improve the quality of our work. We hope that these revisions are satisfactory and that the revised version will be acceptable for publication in Geoscientific Model Development. Thank you very much for your work concerning my paper.

Wish you all the best!

Yours sincerely,

Daichun Wang and Wei You

11/23/2021

---

## Author Comment (AC4)

**Responses to the comments of Reviewer #3:**

We are truly grateful to yours' positive comments and thoughtful suggestions. Those comments are all valuable and very helpful for revising and improving our paper, as well as the important guiding significance to our researches. Based on these comments and suggestions, we have studied carefully and have made correction which we hope meet with approval. All changes made to the text are marked in yellow color. Below you will find our point-by-point responses to the reviewers' comments/ questions:

**General Comments:**

*1. The model resolution, meteorological conditions, and emission data could be other important sources of uncertainty in the air pollution modeling and in fact some of them can be identified in the diagram you show in Fig. 2. However, they are not discussed in the manuscript. Would you be able to quantify these uncertainties in relation to the impact of aerosol field initialization (DA) based on the design of model experiment? For instance, you may consider conducting additional experiment which assimilate meteorological states and aerosol to explore their relative impacts on the subsequent forecast.*

**Response:**

We really appreciate your valuable suggestion. Discussing various sources of uncertainty in the air pollution modeling is of significance, however, this manuscript presented a new development of aerosol optical properties data assimilation (independent developed), which is coupled with the MOSAIC scheme for the first time and different from the GSI tool, so a validation of the developed assimilation system using Himiwari-8 AOT observations was focused in the study. Quantifying these uncertainties may need well-designed model experiments, which would be carried out in the following researches.

We are sorry to say that the developed assimilation system has no capacity of assimilating meteorological data, namely, it only aims at improving aerosol initial conditions. Nevertheless, it can assimilate a wide range of aerosol observations, including total aerosol ($PM_{2.5}$, $PM_{10}$) or component mass concentration, optical properties such as AOD, extinction and backscatter profiles, and attenuated backscatter profile, which would advance aerosol data assimilation. Moreover, we will develop meteorological and aerosol coupling DA methods in the future.

*2. The under-utilization of Himawari-8 AOT product (hourly data) in the context of*

*assimilation frequency (24 h) seems to be obvious. I imagine a strategy with more*

*frequent assimilations could be a unique point to make in this research as the*

*geostationary satellite product used here has such a high temporal resolution.*

*Nevertheless, the relevant discussion is not covered in the manuscript. I would*

*suggest adding more content to address this comment.*

**Response:**

We really appreciate your suggestion. Himawari-8 level 3 AOT_Merged, an improved hourly product, which is derived from level 2 AOT retrievals at a 10 min interval, was employed to conduct assimilation experiments. A daily assimilation frequency seems to be an underutilization of Himawari-8 observations in comparison to its high temporal frequency. Since AOT observations are retrieved at the visible and infrared bands, observations between 0300 and 0800 UTC in the daytime are available for China. In fact, AOT observations are noticeably noisy, which will have a greatly negative impact on assimilation results. Moreover, observations at afternoon are much noisier than those in the morning. For example, surface $PM_{2.5}$ concentration and original (not thinned) Himawari-8 AOT observations at 0300 UTC and 0600 UTC on

25 November 2018 are plotted in Fig. 1 and Fig. 2, respectively. Overall, surface $PM_{2.5}$

mass concentrations change little even with a small decrease at some areas from 0300 to

0600 UTC (Fig. 1b, Fig. 2b) while there is a remarkably increase in AOTs during the same period (Fig. 1a, Fig. 2a). In terms of $PM_{2.5}$, the noticeably increase in AOT

observations should not be considered as normal changes of aerosol but much noise. As a result, more frequent assimilation of AOT observations like this will certainly result in a dramatic overestimation of $PM_{2.5}$ mass concentrations. In terms of evaluation with

$PM_{2.5}$ mass concentration observations, AOT observations at 0300 UTC without no temporal collocation were only assimilated in this study to test the developed assimilation system. As known, data assimilation serves only as a mathematical approach on how to introduce observations into the model, and then improves model initializations and forecasts. Assimilation results are largely determined by observational data, as for how to deal with those with high noise and improve the quality, more researches are needed in the future.

[Figure]

**Figure 1. Observations of the original (not thinned) Himawari-8 AOTs (a) and surface $PM_{2.5}$ mass concentration (b) in D02 at 0300 UTC on 25 November 2018.**

[Figure]

**Figure 2. Same as Fig.1, but at 0600 UTC on 25 November 2018.**

*3. Despite the vertical profiles of background error STDs and auto-correlations are*

*given, the analyzed increments of each aerosol state variables are not seen*

*anywhere in the document. Since the AOD is obtained through the integration of*

*aerosol properties in the atmospheric column, it would be useful to show analyzed*

*results in terms of their vertical distributions and further discuss how would that*

*contribute to the uncertainty of simulation.*

**Response:**

We really appreciated and followed the suggestion. The assimilation process directly produces the analysis increments of 20 aerosol state variables, so it is natural to give the analyzed increments of each aerosol state variable. The analyzed $PM_{2.5}$

increments were computed based on those of each variable and given in Fig. 9 in light of comparing with $PM_{2.5}$ observations (no aerosol state variable observations are available at present). Actually, the increment of each variable contributes greatly to the total $PM_{2.5}$ increment and differs significantly according to its background error

STD. In general, the variable with a larger background error STD has a larger increment and vice versa. Of all state variables, SSN2 has the greatest background error STD, its increment in case of November 25, 2018 is shown in Fig. 3 here, which is similar to that of $PM_{2.5}$.

[Figure]

**Figure 3. Spatial distribution of SSN2 in the background field (a) and analysis (b) as well as the increment (c) in D02 at 0300 UTC on 25 November 2018, these quantities are in unit of ug m⁻³.**

As you mentioned, it would be useful to show vertical distributions of the analyzed increments. Similarly, we has added the vertical distribution of $PM_{2.5}$

analyzed increment, which is shown in Fig. 10 in the revised manuscript (here is shown in Fig. 4), helping to demonstrate the impacts of AOD assimilation on aerosol vertical distributions. And the following information has also been added in the revised manuscript (L670-681). "Since AOD is an atmospheric column measurement, it naturally includes the information of aerosol vertical distributions. Consequently,

AOT assimilation can improve aerosol vertical distributions as well. A vertical cross-section of $PM_{2.5}$ at 0300 UTC on 25 November 2018 is shown in Fig. 10, this cross-section is through Tianjin (marked by the black triangle in Fig. 9). Similar to surface $PM_{2.5}$, suspended $PM_{2.5}$ mass concentrations in the upper air are also enlarged with a wide range from the ground to about 1 km by significantly positive increments generated by assimilation (Fig. 10c). In spite of no observational $PM_{2.5}$ profiles to compare, the vertical distribution in analyses is believed to be closer to the real in terms of the ground $PM_{2.5}$ level (Fig. 10b). It should be noted that the vertical increments are determined by the background error vertical correlation. In a summary,

AOD assimilation is certainly helpful to improve the three-dimensional structures of

$PM_{2.5}$."

[Figure]

**Figure 4. Vertical cross-section of PM₂.₅ in the background field (a) and analysis (b) as well as the increment (c) in D02 at 0300 UTC on 25 November 2018.**

**Specific Comments:**

*4. L32: It is mentioned here that the developed DA system is able to assimilate lidar-based aerosol profiles. However, I did not find any relevant description with respect to the treatment in the followed sections. Would you clarify this?*

**Response:**

We really appreciate your question. Developing a new aerosol data assimilation system, especially for variational method to assimilate unconventional observation data (such as aerosol optical data sources), is a challenging work. Based on the 3DVAR principle, the observation operator determines what type of observations can be assimilated, that is, you need to design and construct the operator according to the observations which will be assimilated. In fact, various aerosol optical properties can be simultaneously calculated through the previous same steps, for example, the process from the size parameter, complex refractive, and aerosol number to optical properties such as extinction and backscatter coefficients, go further, AOD and attenuated backscatter can be computed using extinction and backscatter. In the data assimilation system, these optical quantities have individually corresponding observational data interface. What type of observations are inputted, the assimilation system run corresponding program codes, and this design is easily implemented in practical coding.

For example, if extinction and backscatter profiles are to be assimilated, then the terms in the cost function and its gradient associated with the following AOD and attenuated backscatter are no longer computed. It is worth mentioning that only AOD observations are employed to test the developed assimilation system in this study, so any relevant descriptions of lidar-based extinction or backscatter profiles assimilation are not given.

We will combine assimilate more data sources including surface PM data, satellite derived AOD, attenuated backscatter et al in the near future.

*5.  L237-240: Have you conducted any experiment to test how sensitive this constant*

*error is?*

**Response:**

We really appreciate your question. We have not conducted any experiment to test how sensitive the observation error is. The development and validation of the assimilation system are focused in this study. The observation error plays an important role in the assimilation process, however, it is very difficult to accurately determine it and usually determined based on experience (or tuning parameters).

*6.  L260: Can you give an example of the minimization process, such as reduction of*

*cost function in function of iteration numbers?*

**Response:**

We really appreciate you question. The minimization process is to find the minimum solution to the cost function, which usually employs the descent algorithm, such as the L-BFGS algorithm here which is a limited memory quasi-Newton method for    large    scale    unconstrained    optimization    and    available    at http://users.iems.northwestern.edu/~nocedal/lbfgs.html. In general, the minimization process is a process of iteratively updating control variables. At first, the cost function and its gradient are computed with an initial value of control variables, and the function and gradient values along with control variable values are put into the descent algorithm to obtain a new value of control variables. Then come to the next step, new values of the function and its gradient as well as control variables are altogether put into the descent algorithm again to update the value of control variables, go on like this. The process ends until the convergence condition (the gradient is equal to 0 in theory) is meet or iteration number for example 50 is reached. In the minimization process, the cost function keep reducing, and the reduction is fast in the beginning while it becomes slowly lately. Further more, the reduction depends on the case and is hard to describe in function of iteration numbers. In our study, the max number of iterations is set to 50.

The number of iterations varies with experimental cases.

*7. L288-289: Please include references to supplement statement here*

**Response:**

Done. The following reference has been added: (L293)

Barnard, J. C., Fast, J. D., Paredes-Miranda, G., Arnott, W. P., and Laskin, A.:

Technical Note: Evaluation of the WRF-Chem "Aerosol Chemical to Aerosol Optical

Properties" Module using data from the MILAGRO campaign, Atmos. Chem. Phys.,

10, 7325–7340, https://doi.org/10.5194/acp-10-7325-2010, 2010.

*8. L291: Should be black car"b"on and organic car"b"on*

**Response:**

Done. (L294-295)

*9. L369: Would this introduce any inconsistency between nonlinear model and TL?*

*Also, I am curious how did you deal with if statements in the code if there's any.*

**Response:**

We really appreciate your question. The Optical Module within WRF-Chem is a developed routine package, it can compute a large number of aerosol optical quantities, such as aerosol scatter phase functions. However, these codes have nothing with the development of the assimilation system. Thus, when transplanting the Optical Module to establish the observation operator, these irrelevant codes should be removed to reduce the difficulty in tangent linear (TL) and adjont (AD) coding. Also, above-mentioned process can improve computing efficient.

The conditional statements remain unchanged when establishing the TL or AD codes of if statements. TL or AD codes of the assignment statements are needed to add into if statements. TL statements are arranged in the same order as assignment statements, but AD statements are arranged in a reverse order.

*10. L389: Since this manuscript documents the development of a DA package, it is of necessity to show the result of TL/AD test. For example, it is common to show the plot of gradient check with respect to various orders of perturbation.*

**Response:**

We really appreciated your suggestion. TL/AD test is necessary for establishing TL and AD codes, which only serves as the validation of the codes after all it is a huge work to finish the TL/AD codes and easy to make mistakes, so it seems unnecessary to give the result of TL/AD test in the manuscript. The following table (Tab. 1) shows the gradient with respect to perturbations in both directions. It is noted that initial perturbations are set to 20 and -20, respectively, and the gradient (radio) of AOD with respect to control variables was calculated by halving the perturbation every time. Eventually, the gradient approaches 1 in both directions.

**Table 1.    TL/AD test results**

| number | positive perturbation | ratio (gradient) | negative perturbation | ratio (gradient) |
|--------|----------------------|------------------|----------------------|------------------|
| 1 | 20.00000 | 1.02831070096536 | -20.00000 | 0.995594423135122 |

| 2 | 10.00000 | 1.02728481026492 | -10.00000 | 0.997059224601074 |
|---|---|---|---|---|
| 3 | 5.000000 | 1.02644276988709 | -5.000000 | 0.997750286836985 |
| 4 | 2.500000 | 1.02579561769594 | -2.500000 | 0.998080650773033 |
| 5 | 1.250000 | 1.02542213463021 | -1.250000 | 0.998239359890258 |
| 6 | 0.6250000 | 1.02522400926412 | -0.6250000 | 0.998316741235688 |
| 7 | 0.3125000 | 1.02512225357477 | -0.3125000 | 0.998354903969795 |
| 8 | 0.1562500 | 1.02507072260859 | -0.1562500 | 0.998373850019414 |
| 9 | 7.8125000E-02 | 1.02504479642776 | -7.8125000E-02 | 0.998383288869707 |
| 10 | 3.9062500E-02 | 1.02503179348556 | -3.9062500E-02 | 0.998387999717800 |
| 11 | 1.9531250E-02 | 1.02502528213119 | -1.9531250E-02 | 0.998390352987688 |
| 12 | 9.7656250E-03 | 1.02502202388487 | -9.7656250E-03 | 0.998391529132607 |
| 13 | 4.8828125E-03 | 1.02502039438236 | -4.8828125E-03 | 0.998392116963912 |
| 14 | 2.4414062E-03 | 1.02501957932535 | -2.4414062E-03 | 0.998392411082556 |
| 15 | 1.2207031E-03 | 1.02501917199313 | -1.2207031E-03 | 0.998392557990852 |

*11. L418: Please cite this reanalysis product and provide the link of the data source.*

**Response:**

Done. We have added the link of the data source (L424-425).

*12. L422: The assimilation cycle time (24 hours) seems to be coarse in relation to data availability. Please discuss how it is designed and clarify if there's any limitation on the data coverage or quality, etc.*

**Response:**

We really appreciate your question. As discussed above, Himawari-8 level 3 observations between 0300 and 0800 UTC in the daytime are available for China. AOT observations are noticeably noisy, which will have a greatly negative impact on assimilation results. In terms of $PM_{2.5}$, directly assimilating AOT with noises will result in a dramatic overestimation of $PM_{2.5}$ mass concentrations. The 24 h assimilation frequency was designed only to test the developed system.

*13. L424-426: The statement here is contradictory to the design of assimilation cycles.*

*Please explain.*

**Response:**

We really appreciate your question. As explained above, more frequent assimilation of AOT observations with much noise will cause the significant overestimation of $PM_{2.5}$ mass concentrations. Nevertheless, In terms of evaluating with AOT observations, more frequent assimilation may have better effects.

*14. L441: I am not sure this is the best treatment as it could further smooth out the*

*observed data. Please address.*

**Response:**

We really appreciate your question. We aggregated AOT observations in the spatial resolution of the model, which is also employed by other researchers (Yumimoto et al., 2016; Dai et al., 2019; Ha et al., 2020). How to treat the dataset with a high spatial resolution before assimilation may need further researches. We aggregated AOT observations by averaging them in one grid cell so that the resolution of them matches that of the model, smoothing out the observed data, however, this approach can filter out much noise to improve the quality.

*15. L443 and L463: Fig. 3b is mentioned earlier than Fig. 3a. I would suggest*

*swapping them for the fluency of reading.*

**Response:**

We followed the suggestion. Fig.3b and Fig. 3a have been swapped in the revised manuscript (L451, L472).

*16. L492: It looks like the similar DA procedure is also carried out over the D01 but at*

*least with different treatment in data thinning. Have you done any experiment*

*without assimilation in D01? If true, what was the impact of additional DA in D01.*

**Response:**

We really appreciated your question. A two-level nested domain configuration was employed to run simulation experiments. The outer domain D01 is at a horizontal resolution of 27km, and the inner domain D02 is at a resolution of 9km. The AOT

observations are thinned using D01 grid and D02 grid, respectively. The same assimilation procedure was carried out over D01 and D02, separately, but with data of different resolutions, to improve individual aerosol initial conditions. In the control experiment, both D01 and D02 simulations were performed without assimilation. The

D02 simulations were only evaluated with various observations and the evaluation was shown in this study because the AOT observations are mainly distributed in D02.

Of course, we can evaluate the impacts of D01 assimilation on D01 simulations as well, nevertheless, it seems a repeated work in terms of testing the development.

*17. L532: Is it possible to estimate the correlation length with the observational data or*

*alternatively the analysis after assimilation?*

**Response:**

We really appreciated your question. It is a good idea that using the analysis after assimilation to estimate the correlation length. We will conduct the test in the future.

*18. L577-579: Sentences such as these in the manuscript could be trimmed to shorten*

*the length.*

**Response:**

We followed your valuable suggestion. The relevant sentences have been revised as "The higher scores of the metrics CORR, RMSE, and BIAS would demonstrate the better assimilation performance and vice versa" (L586-587).

*19. L587: Please elaborate more on this. Would the uncertainty mostly be on the*

*magnitude or something else?*

**Response:**

We really appreciated your suggestion. AOD simulation was performed at a wavelength of 500 nm, the same as Himawari-8 AOT retrievals, whereas MODIS

AOD is retrieved at 550 nm. It is obvious that the difference in the wavelength (500nm and 550nm) would affect the evaluation when evaluating the AOD simulation with MODIS AOD, however, the evaluation is convincing because the wavelength difference is minor.

*20. L606: The red triangles in Fig. 1b are hardly distinguished from one another as*

*they are basically overlapped with each other. Please try to make them more visible.*

*Add another zoomed-in map may help achieve that.*

**Response:**

We have added a zoomed-in map as Fig. 1c for AERONET sites in Beijing area in the revised version, which is also given as Fig. 5 below:

[Figure]

**Figure 5. A zoomed-in map for AERONET sites in Beijing area, including Beijing, Beijing-CAMS,**
**Beijing_PKU, Beijing_RADI, XiangHe.**

*21. L610: What is the temporal resolution of AERONET observations? From the time*

*series plot of Fig. 8, it looks like the data is mostly only available around 00 UTC of*

*each day.*

**Response:**

We really appreciate your question. The temporal resolution of AERONET

observations is several minutes, and the data in the daytime is only available because sun photometer measurements of the direct solar radiation is used to retrieve AOD.

*22. L615-616: Any explanation why model has worse skill at XuZhou-CUMT? It seems*

*the event on Nov. 25 is more severe than Nov. 26 at this site and not captured as*

*well.*

**Response:**

We are so sorry to give a rational explanation, the worse model skill at

XuZhou-CUMT is probably due to emissions, which is needed to further study.

*23. L617-618: Any guess on this? Have you looked at the meteorological conditions on*

*these days? Could it be associated with the intensity of wind speed?*

**Response:**

We are so sorry that we have not looked at the meteorological conditions on these days, and studied the impacts of them on assimilation. The intensity of wind speed has actually an important impact on assimilation, so combined assimilation of meteorological and aerosol states should be performed in the future.

*24. L622: It would be easier for reader to understand if the data distribution map of*

*Nov. 26 is also provided. Along the same line, I would suggest adding information*

*of available data amount in Fig. 8 to address this.*

**Response:**

We really appreciate your suggestion. The AOD data amount has a significant impact on assimilation, for example, no available AOD data shown in Fig. 6a can be assimilated in Beijing area due to cloud contamination where a more severe pollution happened on 26 November 2018 shown in Fig. 6b so that no assimilation benefits are generated to improve aerosol forecasts in Beijing area, meaning the control experiment and assimilation experiment on 26 November 2018 have the same performance (shown in Fig. 8a, 8b, 8c, 8d, 8e in the manuscript). The available data amount is variable from

23 to 29 November 2018. What is more, the amount of data is same, the assimilation effect may differ greatly due to different pollution cases.

[Figure]

[Figure]

**Figure 6. Observations of the thinned Himawari-8 AOTs (a) and surface PM₂.₅ mass concentration (b) in D02 at 0300 UTC on 26 November 2018.**

*25. L643: You may remove "between analyses and the background field" since*

*increment has been defined in the earlier paragraph.*

**Response:**

Done. The words "between analyses and the background field" has been removed in the revised manuscript (L656-657).

*26. L644-645: The of color bar scales in Fig. 3a and Fig. 9 are not consistent, which*

*makes it hard to compare them visually. Please consider modify them.*

**Response:**

Done. We have modified the color bar scales in Fig. 9.

*27. L645: Need to mark where Tianjin is in the map, otherwise one may not know which*

*location you talked about.*

**Response:**

    Done. We have marked Tianjin with a small black triangle in the map (L659).

*28. L644: Panels in Fig. 9 are not sufficient to conclude the underestimation in control experiment as no observation is provided.*

**Response:**

    We really appreciated your suggestion. Fig. 9a shows surface $PM_{2.5}$ mass concentrations in the background field at 0300 UTC on 25 November 2018, whereas corresponding observations are provided in Fig. 3b.

    We would like to express our great appreciation to you for the valuable and pertinent comment on our manuscript, which is crucial to improve the quality of our work. We hope that these revisions are satisfactory and that the revised version will be acceptable for publication in Geoscientific Model Development. Thank you very much for your work concerning my paper.

    Wish you all the best!

    Yours sincerely,

Daichun Wang and Wei You

11/24/2021

---

## Author Comment (AC5)

**Responses to the comments of Reviewer #4:**

We are truly grateful to yours' positive comments and thoughtful suggestions. Those comments are all valuable and very helpful for revising and improving our paper, as well as the important guiding significance to our researches. Based on these comments and suggestions, we have studied comments carefully and have made correction which we hope meet with approval. All changes made to the text are marked in blue color. Below you will find our point-by-point responses to the reviewers' comments/questions:

**Specific Comments:**

*1. L144 duo to->due to*

**Response:**

Done. (L147)

*2. L291 carton-> carbon*

**Response:**

Done. (L294)

*3. L305 What do you mean distributing the increments using the mass concentration*
*background error STD? Please clarify this.*

**Response:**

We really appreciated your question. The assimilation process will directly generate analysis increments of 20 control variables, however, these control variables are not completely consistent with model variables within MOSAIC. For those consistent with model variables, their increments can be directly used to adjust model variables, while for those lumped control variables, their increments correspond to 2 or 3 model variables, for instance, the control variable SSN1 correspond to 3 model variables, i.e. *so4_a01*, *no3_a01*, and *nh4_a01*, which are sulfate, nitrate, ammonium mass concentrations at the first size bin, respectively, thus, distributing the increment of SSN1 over three model variables *so4_a01*, *no3_a01*, and *nh4_a01* is necessary.

How to distribute? A simple way is to determine the distribution ratio. When estimating background error covariance using the NMC method, we can employ differences between 48 h and 24 h forecasts valid at the same time (i.e. 0000 UTC) for every model variable within a period of one month (November 2018) to set up a sample and figure out the background error standard deviation (STD) in mass concentration. For example, the computed STDs of *so4_a01*, *no3_a01*, and

*nh4_a01*are $c_1$, $c_2$, and $c_3$, respectively, thus, the corresponding distribution ratios are calculated as $c_1/(c_1+c_2+c_3)$, $c_2/(c_1+c_2+c_3)$, $c_3/(c_1+c_2+c_3)$.

*4. L540 You said the vertical correlation of every variable is similar, however, you*

*subsequently said vertical correlations differ among aerosol variables. Please*

*clarify it. Besides, since the AOT observation has no vertical information, how do*

*you assume the vertical information of the AOT observations?*

**Response:**

We really appreciated your question. We said the vertical correlation of every variable is similar, meaning that vertical correlation plots for every variable look similar. Because the vertical correlation describes the auto-correlation between two layers at different heights, the vertical correlation is a symmetric matrix and the maximum 1 is on the diagonal, which is common to all variables. Therefore, the vertical correlation of every variable is similar. However, vertical correlations among aerosol variables are not the same. Given a correlation more than 0.8, some variables have a larger domain while some have a less domain, which indicates that vertical correlations differ among aerosol variables.

AOT is an atmospheric column measurement, it has no vertical information.

When assimilating AOT observations, it does not need to assume the vertical information of the AOT observations.

*5. Fig.7 Can you explain why the assimilation has little effects on the significant*

*underestimates of the AOTs? Such as the observed AOTs are around 1-1.5,*

*whereas the simulated ones are around 0.*

**Response:**

Thank you so much for your question. In general, the assimilation has significant effects on AOT simulation, but has little effects on the some significant underestimates of the AOTs. This phenomenon is probably due to uncertainties in aerosol emissions as well as meteorological boundary conditions. Emission data is another important factor that influences the aerosol simulation. Simultaneous assimilation of aerosol data to updating aerosol emission and initial field may reduce this phenomenon in the future.

We would like to express our great appreciation to you for the valuable and pertinent comment on our manuscript, which is crucial to improve the quality of our work. We hope that these revisions are satisfactory and that the revised version will be acceptable for publication in Geoscientific Model Development. Thank you very much for your work concerning my paper.

Wish you all the best!

Yours sincerely,

Daichun Wang and Wei You

11/24/2021